# Honokiol: A Review of Its Anticancer Potential and Mechanisms

**DOI:** 10.3390/cancers12010048

**Published:** 2019-12-22

**Authors:** Chon Phin Ong, Wai Leong Lee, Yin Quan Tang, Wei Hsum Yap

**Affiliations:** School of Biosciences, Faculty of Health and Medical Sciences, Taylor’s University Lakeside Campus, No. 1, Jalan Taylor’s, Subang Jaya 47500, Malaysia; chonphin96@gmail.com (C.P.O.); leewl0108@gmail.com (W.L.L.)

**Keywords:** honokiol, anticancer, mechanism, signalling pathway

## Abstract

Cancer is characterised by uncontrolled cell division and abnormal cell growth, which is largely caused by a variety of gene mutations. There are continuous efforts being made to develop effective cancer treatments as resistance to current anticancer drugs has been on the rise. Natural products represent a promising source in the search for anticancer treatments as they possess unique chemical structures and combinations of compounds that may be effective against cancer with a minimal toxicity profile or few side effects compared to standard anticancer therapy. Extensive research on natural products has shown that bioactive natural compounds target multiple cellular processes and pathways involved in cancer progression. In this review, we discuss honokiol, a plant bioactive compound that originates mainly from the *Magnolia* species. Various studies have proven that honokiol exerts broad-range anticancer activity in vitro and in vivo by regulating numerous signalling pathways. These include induction of G0/G1 and G2/M cell cycle arrest (via the regulation of cyclin-dependent kinase (CDK) and cyclin proteins), epithelial–mesenchymal transition inhibition via the downregulation of mesenchymal markers and upregulation of epithelial markers. Additionally, honokiol possesses the capability to supress cell migration and invasion via the downregulation of several matrix-metalloproteinases (activation of 5′ AMP-activated protein kinase (AMPK) and KISS1/KISS1R signalling), inhibiting cell migration, invasion, and metastasis, as well as inducing anti-angiogenesis activity (via the down-regulation of vascular endothelial growth factor (VEGFR) and vascular endothelial growth factor (VEGF)). Combining these studies provides significant insights for the potential of honokiol to be a promising candidate natural compound for chemoprevention and treatment.

## 1. Introduction

Cancer is the outcome of rampant cell division which is associated with cell cycle disorganisation [1], leading to uncontrolled cell proliferation. In addition, it also involves the dysregulation of apoptosis, immune evasion, inflammatory responses, and ultimately, metastatic spread [2]. Over the last few decades, our progressive understanding of the aetiology of cancer together with advancement of cancer treatment, detection, and prevention, have contributed towards receding cancer mortality around the world [3]. However, more than half of cancer cases were diagnosed at a later stage of cancer progression [4]. According to a study by Bray et al. [5], the worldwide estimated number of new cancer cases for the year 2018 was 18.1 million in both sexes and across all ages. Amongst all the cancer types, lung, breast, and colorectum have topped the charts with approximately 2.1 million, 2.1 million, and 1.8 million cases, respectively. On the other hand, the estimated number of deaths was approximately 9.6 million. Asia accounted for more than half of the cancer deaths (57.3%), followed by Europe (20.3%), and America (14.4%). Lung cancer has caused the highest number of deaths due to substandard prognoses. Attempts to develop the effective prevention of cancer may diminish the incidence rate for some cancers, for instance lung cancer in North America and Northern Europe. These western countries have implemented tobacco control in order to avert involuntary exposure to tobacco and minimise active smoking within the community. Unfortunately, a majority of the population are still facing an upsurge of cancer diagnosis, demanding treatment and care [5].

The common treatment regimens for cancer patients include surgery, chemotherapy, and radiotherapy [6]. Although some of these regimens represent the first-in-line options for cancer treatment, the lack of selectivity towards neoplastic cells and the development of drug toxicity has caused these therapeutic effects to recede slowly, rendering it ineffective over the years [7]. Additionally, multidrug resistance tumours pose a severe threat and have been responsible for numerous cancer-related deaths [8]. A modern approach to target multiple cell regulating pathways is mandatory in order to provide highly efficient and targeted cancer therapy. For instance, combination therapy that targets different pathways exhibit significantly lower toxicity towards normal cells compared to mono-therapy [9]. Currently, the development of anticancer drugs possessing the capability to overcome common mechanisms of chemoresistance with minimal toxicity effects would be considered a breakthrough in cancer research [2].

Approximately 70–95% of the world population continues to use traditional medicinal herbs, plants, and fruits which contain valuable bioactive compounds with therapeutic effects to maintain health, as well as to prevent or treat physical and mental illnesses [10]. These biologically active compounds provide extensive opportunities in uncovering competent anticancer agents [2,11]. A majority of the anticancer drugs that are currently in use originate from plants, marine organisms, and microorganisms, such as the well-known plant-derived anti-cancer drugs Paclitaxel (Taxol^®^) and Camptothecin (CPT) [12].

The *Magnolia* genus is widely distributed throughout the world, especially in East and South-East Asia [13]. Among the *Magnolia* species, *Magnolia officinalis* and *Magnolia obovata* are commonly used in traditional Chinese (known as “Houpu”) and Japanese herbal medicine [13,14]. The traditional prescriptions named Hange-koboku-to and Sai-boku-to, which contain the *Magnolia* bark, are still used in modern clinical practice in Japan [15]. There are several potent bioactive compounds in the *Magnolia* species have been identified including honokiol, magnolol, obovatol, 4-*O*-methylhonokiol, and several other neolignan compounds [13,15,16]. This paper highlights the potential anticancer effect of a simple biphenyl neolignan found in this *Magnolia* family, namely honokiol.

Honokiol was traditionally used for anxiety and stroke treatment, as well as the alleviation of flu symptoms [14]. In recent studies, this natural product displayed diverse biological activities, including anti-arrhythmic, anti-inflammatory, anti-oxidative, anti-depressant, anti-thrombocytic, and anxiolytic activities [13,14,16]. Furthermore, it was also shown to exert potent broad-spectrum anti-fungal, antimicrobial, and anti-human immunodeficiency virus (HIV) activities [13]. Due to its ability to cross the blood–brain barrier, honokiol has been deemed beneficial towards neuronal protection through various mechanism, such as the preservation of Na^+^/K^+^ ATPase, phosphorylation of pro-survival factors, preservation of mitochondria, prevention of glucose, reactive oxgen species (ROS), and inflammatory mediated damage [17]. Hence, honokiol was described as a promiscuous rather than selective agent due to its known pharmacologic effects. Recent studies have been focused on the anti-cancer properties of honokiol, emphasising its tremendous potential as an anticancer agent. In this review, we summarise the anti-cancer properties of honokiol, together with its mechanism of action, based on in vitro and in vivo experimental evidence. In addition, we also summarize the current data on its pharmacological relevance and potential delivery routes for future applications in cancer prevention and treatment.

## 2. Research Methodology

A systematic search was performed to identify all relevant research papers published on the use of honokiol as a potent anticancer treatment using PubMed (1994–present) and Web of Sciences (1994–present). The search strategy was performed using several keywords to track down the relevant research articles including ‘honokiol’, ‘cancer’, ‘cancer statistics’, ‘structural’, ‘metabolites’, ‘mechanism’, ‘cell death’, ‘apoptosis’, ‘anti-inflammatory’, ‘anti-tumour’, ‘antioxidant’, ‘cell proliferation’, ‘cytotoxicity’, ‘cell cycle arrest’, ‘metastasis’, ‘tumour’, ‘angiogenesis’, ‘absorption’, ‘metabolism’, ‘toxicity’, ‘distribution’, ‘elimination’, ‘solubility’, ‘nanoparticles’, and ‘delivery’.

## 3. Structure Activity Relationship and Its Derivatives

Honokiol bioactive compounds are easily found in the root and stem bark of the *Magnolia* species, although some studies have also found them in seed cones [13,18]. Due to the structural resemblance of both honokiol and magnolol in the *Magnolia* bark, the extraction of pure honokiol and magnolol cannot be achieved using conventional column chromatography nor thin-layer chromatography. Eventually, their purification process requires a costly alternative like electromigration [16]. The only difference between honokiol and magnolol in terms of structure is only in the position of the hydroxyl group, as shown in Figure 1. In 2007, Chen et al. developed a rapid separation technique using high-capacity high-speed counter-current chromatography (HSCCC) to isolate and purify honokiol and magnolol from crude extracts of *Magnolia* plants. Within 20 min, the resulting fraction has a purity of 98.6% honokiol, indicating that this method exhibited substantial efficiency in honokiol extraction [19]. Two years later, another team of researchers formulated a time-effective synthetic method while providing higher yielding honokiol using Suzuki-Miyaura coupling and Claisen rearrangement as key steps of the synthetic pathway of honokiol. The five steps of the honokiol synthesis pathway includes bromination, Suzuki coupling, allylation, one-pot Claisen’s rearrangement, and demethylation, eventually resulting in a 32% overall yield [20]. The emergence of the synthetic method for honokiol has alleviated the risk of extinction of the *Magnolia* species.

Natural bioactive compounds often serve as lead templates and are subjected to structural modification to improve pharmacological activity, physiochemical properties, along with pharmacokinetics, to generate clinically useful structures [21]. According to Anand et al. [22], a comprehensive study of the natural and synthetic analogues of a drug molecule is crucial to determining its fundamental pharmacophores. As seen in Figure 1, honokiol contains two phenyl rings substituted with hydroxyl and allyl groups. In a study conducted by Bohmdorfer et al. [23], it was found that the predominant metabolic pathways of honokiol in the human liver was through sulfation and glucuronidation (Phase II metabolism) of the free hydroxyl groups, inducing rapid excretion and shortening its half-life [23]. Moreover, Lin et al. [24] have hypothesised that the hydroxyl groups on the biphenyl skeleton of honokiol could be subjected to metabolic oxidation by Phase I enzymes, thus diminishing its efficacy.

**Figure 1 cancers-12-00048-f001:**
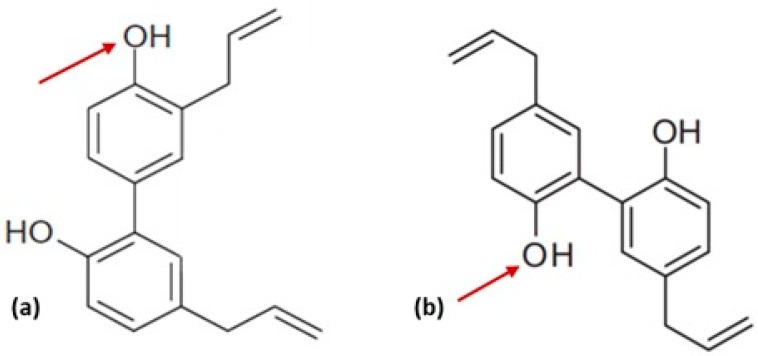
(**a**): The structure of honokiol [24]; (**b**): The structure of magnolol [25]. Arrow indicates the difference in the position of hydroxyl group between honokiol and magnolol.

Through the alteration of the top and bottom rings by changing the substitution pattern at its bottom ring and replacing the hydroxyl group in the top ring with a methoxy group, six different analogues were produced, as shown in Figure 2. A structure–activity relationship (SAR) study was conducted and it was found that replacing the hydroxyl group in the top ring of honokiol with a methoxy group greatly improved its cytotoxicity against lung, melanoma, and colon cancer cells. The two hydroxyl group substituted analogues (3′-Bromo-3,5′-dy-allyl-2′hydroxyl-4-methoxy-1,1′-biphenyl and 3,3′-Diallyl-4-methoxy-4′-hydroxy-1,1′-biphenyl) have induced G0/G1 phase cell cycle arrest and a swift decrement in Cdk1 and cyclin B1 protein levels, similarly to the parental honokiol compound [24]. Overall, obstruction of the potential oxidation of the phenolic hydroxyl group in the biphenyl group skeleton of honokiol improved its anti-cancer effect.

## 4. Anticancer Properties of Honokiol

### 4.1. In Vitro Studies

Honokiol has been shown to exhibit antiproliferation effects against numerous cancer cells, including bone, bladder, brain, breast, blood, and colon, as shown in Table 1. Generally, the concentrations used for the in vitro studies are between 0–150 μM, which majority of these concentration ranges have been shown to significantly inhibit cell proliferation or cell viability of various cancer cell lines. The trend for the IC_50_ values of numerous cancer cell lines were time-dependent, whereby the IC_50_ values decreases as duration of the experiment increases. As seen in Table 1, human blood cancer Raji cells were highly susceptible to honokiol treatment (IC_50_ = 0.092) compared to highly resistant human nasopharyngeal cancer HNE-1 cells (IC_50_ = 144.71 μM). Interestingly, honokiol has been shown to exhibit minimal cytotoxicity against on normal cell lines, including human fibroblast FB-1, FB-2, Hs68, and NIH-3T3 cells [25,26,27,28]. The low cytotoxicity of honokiol treatment against normal cell lines should be emphasised as current chemotherapeutic regimens have a considerable amount of side effects that harm cancer patients.

Many chemotherapeutic agents have been shown to induce severe systemic toxicity and several side effects due to their deficient pharmacokinetic profiles and non-specific distribution in the body [29]. In Yang et al.’s study [30], they have encapsulated honokiol into nanopolymers to enhance its permeability and specificity against cancer cells. They utilised the active targeting nanoparticles-loaded honokiol (ANTH) in their in vitro studies against human nasopharyngeal cancer HNE-1 cells, and this incorporation exhibited significantly lower IC_50_ values compared to free honokiol treatment. As a result, the incorporation or encapsulation of honokiol in transporting vehicles can improve the anticancer effects and concurrently overcome the water solubility issue of honokiol itself. This has shown to be a promising regimen for anticancer treatment in the future.

Furthermore, it is worthy to note that honokiol can enhance the antineoplastic effects of several chemotherapeutic agents when cells are treated in combination treatment of both honokiol and the chemotherapeutic agent. In Wang et al.’s study [31], they have shown that honokiol has enhanced the in vitro cytotoxicity of paclitaxel against human cervix cancer cell lines. The combination treatment has resulted in approximately 10–60% increase of apoptotic cells and inhibition of cell viability when compared to honokiol treatment alone [31]. In another study, honokiol potentiated the apoptotic effect of both doxorubicin and paclitaxel against human liver cancer HepG2 cells. Honokiol enhanced the apoptotic effects of paclitaxel and doxorubicin by 22% and 24% respectively [32].

**Table 1 cancers-12-00048-t001:** The anticancer effects of honokiol against cancer cells in in vitro experiments.

Cell Lines		Mechanism of Action	Concentration Used	Efficacy/IC_50_ (Exposure Time)	References
Colorectal cancer	RKO	Inhibit cell proliferation Induce G1 phase cell cycle arrest Induce apoptosis ↓ Bcl-xL; ↑ Caspase-3 & caspase-9	0–150 μM	46.76 μM (68 h)	[33]
	HCT116, HCT116-CH2, HCT116-CH3	Inhibit cell proliferation Induce G0/G1 & G2/M phase cell cycle arrest: ↓ cyclin D1 & A1; ↑ p53 phosphorylation Induce apoptosis: ↓ Caspase-3; ↓ Bcl-2; ↑ Bax protein	25 μM Honokiol with 2.5 or 5.0 Gy IR	N/A	[34]
	HT-29	Inhibit cell growth & proliferation Induce G1 phase cell cycle arrest: ↓ Cdk1 & cyclin B1	0–50 μM followed by 0–5 Gy IR	23.05 μM (24 h) 13.24 μM (72 h)	[24]
	HCT116 & SW480	Inhibit cell proliferation via Inhibition of Notch signalling: ↓ Notch1 & Jagged-1; ↓ Hey-1 & Hes1; ↓ γ-secretase complex; ↓ Skip1 Induce apoptosis: ↑ caspase-3/-7 activity; ↓ Bcl-2 & Bcl-xL; ↑ Bax protein; ↓ cyclin D1 & c-Myc; ↑ p21^WAF1^ protein Inhibit primary and secondary colonosphere formation	0–50 μM	N/A	[35]
	RKO & HCT116	Inhibit cell viability Induce apoptosis: ↑ caspase-3, caspase-8 & caspase-9 activation; ↑ DR5 & cleaved PARP proteins; ↑ survivin protein; ↑ phosphorylated p53 & p53 proteins; ↓ PUMA protein	0–60 μM	RKO: 38.25 μM (24 h) HCT116: 39.64 μM (24 h)	[36]
Blood cancer	B-CLL	Inhibit cell viability Induce apoptosis: ↑ caspase-3 activity; ↑ caspase-8 & caspase-9 activation; ↓ caspase-9; ↑ Bax protein; ↓ Mcl-1 protein	0–100 μM	49 μM (6 h) 38 μM (24 h)	[37]
	Raji, Molt-4	Inhibit cell growth: ↓ p65; ↓ NF-κB Induce apoptosis: ↑ JNK activation Increase ROS activity: ↑ Nrf2 & c-Jun protein activation	0–2.5 μM	Raji: 3.500 μM (24 h) 0.092 μM (72 h) Molt-4: 0.521 μM (24 h)	[38]
Breast cancer	MCF-7, MDA-MB-231, SKBR-3, ZR-75-1, BT-474	Inhibit cell viability and growth: ↓ EFGR; ↓ MAPK/PI3K pathway activity Induce apoptosis: ↑ PARP protein degradation; ↓ caspase-8; ↑ Bax proteins Induce G1 phase cell cycle arrest: ↓ cyclin D1; ↑ p21 & p27	0–100 μM	MCF-7: 40 μM (24 h) MDA-MB-231: 33 μM (24 h) SKBR-3: 29 μM (24 h) ZR-75-1: 39 μM (24 h) BT-474: 50 μM (24 h)	[39]
	MCF-7, MDA-MB-231	Inhibit cell clonogenicity Inhibit cell anchorage-dependent colony formation Inhibit cell growth, migration & invasion: ↓ pS6K & 4EBP1 phosphorylation; ↑ AMPK activation; ↓ mTORC1 function; ↑ LKB1 & cytosolic localisation	1–25 μM	N/A	[40]
	MCF-7, MDA-MB-231, SUM149, SUM159	Inhibit cell migration & invasion: ↑ AMPK phosphorylation; ↑ LKB1 Inhibit stem-like characteristics: ↓ Oct4, Nanog & Sox4 protein; ↓ STAT3; ↓ iPSC inducer mRNA	5 μM	N/A	[41]
	MCF-7, MDA-MB-231, T47D, SKBR-3, Zr-75, BT-474	Inhibit cell growth: ↓ PI3K/Akt/mTOR signalling Inhibit cell invasion Induce G0/G1 phase cell cycle arrest: ↓ cyclin D1 & cyclin E; ↓ Cdk2 & c-myc; ↑ PTEN Induce apoptosis: ↑ caspase-3, caspase-6 & caspase-9 activation	0–40 μM	MCF7: 34.9 μM (24 h) 13.7 μM (48 h) 13.5 μM (72 h) 10.5 μM (96 h) MDA-MB-231: 56.9 μM (24 h) 44.4 μM (48 h) 16.0 μM (72 h) 12.0 μM (96 h) T47D: 47.7 μM (24 h) 41.6 μM (48 h) 17.6 μM (72 h) 7.1 μM (96 h) SKBR-3: 76.1 μM (24 h) 68.1 μM (48 h) 62.7 μM (72 h) 15.7 μM (96 h) ZR-75: 71.1 μM (24 h) 58.1 μM (48 h) 28.7 μM (72 h) 14.5 μM (96 h) BT-474: 80.2 μM (24 h) 65.6 μM (48 h) 39.5 μM (72 h) 15.1 μM (96 h)	[42]
	MDA-MB-231	Inhibit cell proliferation: ↓ c-Src/EGFR-mediated signalling pathway; ↓ c-Myc protein Induce G0/G1 phase cell cycle arrest: ↓ cyclin A, cyclin D1 & cyclin E; ↓ Cdk2, Cdk4 & p-pRb^Ser780^; ↑ p27^Kip−1^ Induce apoptosis: ↑ caspase-3, caspase-8 & caspase-9 cascade; ↓ Bcl-2 & Bid protein; ↑ PARP cleavage	0–100 μM	59.5 μM (72 h)	[43]
Lung cancer	A549	Inhibit cell growth & proliferation Induce G0/G1 phase cell cycle arrest: ↓ Cdk1 & cyclin B1	0–50 μM	12.51 μM (24 h) 7.75 μM (72 h)	[24]
	A549, H460, H226, H1299	Reduce invasive potential Inhibit PGE_2_-induced cell migration: ↓ PGE_2_ production ↓ COX-2 ↑ β-catenin degradation ↓ NF-κB/p65 activity ↓ IKKα	0–20 μM	N/A	[44]
	A549, H1299	Inhibit cell viability and growth: ↓ class I HDAC proteins; ↓ HDAC activity; ↑ histone acetyltransferase (HAT) activity; ↑ histone H3 & H4 Induce G1 phase cell cycle arrest: ↓ cyclin D1 & cyclin D2; ↓ Cdk2, Cdk4 & Cdk6	0–60 μM	N/A	[45]
	H460 & A549	Inhibit cell proliferation Induce apoptosis: ↑ cathepsin D; ↑ cleaved PARP; ↑ caspase-3 Inhibit autophagy: ↑ p62; ↑ LC3-II	0–60 μM	H460: ~30 μM (48 h) A549: ~40 μM (48 h)	[46]
	Pc9-BrM3 & H2030-BrM3 (brain metastatic)	Inhibit cell proliferation and cell invasion: ↓ STAT3 protein phosphorylation; ↓ STAT-3 mediated mitochondrial respiratory function	0–50 μM	PC9-BrM3: 28.4 μM (48 h) H2030-BrM3: 25.7 μM (48 h)	[47]
	H23, A549 & HCC827	Inhibit cell growth Induce G1 phase cell cycle arrest: ↓EGFR; ↓ class I HDAC; ↓ class IIb HDAC6 activity; ↑ Hsp90 acetylation & EGFR degradation	0–40 μM	A549: 23.55 μM (24h)	[48]
	H460, A549, H358	Inhibit cell growth: ↓ c-RAF, ERK & AKT phosphorylation Inhibit colony formation capacity Induce apoptosis: ↑ Bax protein; ↓ Bcl-2 protein; ↑ PARP cleavage Induce G1 phase cell cycle arrest: ↓ cyclin D1; ↑ p21 & p27; ↓ P70S6k kinase activity Induce autophagy: ↑ LC3-I conversion to LC3-II; ↑ Sirt3 mRNA & protein; ↓ Hif-1α protein	0–80 μM	H460: 30.42 μM (72 h) A549: 50.58 μM (72 h) H358: 59.38 μM (72 h)	[49]
	A549 & 95-D	Inhibit cell viability Induce apoptosis: ↑ ER stress signalling pathway activation; ↑ GRP78, phosphorylation PERK & phosphorylated IRE1α; ↑ cleaved caspase-9 & CHOP; ↓ Bcl-2 protein; ↑ Bax, caspase-3 & caspase-9 Inhibit cell migration	0–60 μM	N/A	[50]
	CH27, H460 & H1299	Inhibit cell growth Induce apoptosis: ↓ Bcl-XL; ↑ mitochondrial cytochrome c release; ↑ BAD protein; ↑ caspase-1, caspase-2, caspase-3, caspase-6, caspase-8 & caspase-9 activity; ↑ PARP cleavage	0–100 μM	CH27: 40.9 μM (24 h) H460: 41.4 μM (24 h) H1299: 34.7 μM (24 h)	[25]
	MSTO-211H	Inhibit cell viability Induce apoptosis: ↑ PARP cleavage; ↑ caspase-3 activation; ↓ Bid & Bcl-xL protein; ↑ Bax protein; ↓ Mcl-1 & survivin protein; ↓ Sp1 Induce G1 phase cell cycle arrest: ↓ cyclin D1	0–22.5 μM	N/A	[51]
Skin cancer	SK-MEL2 & MeWo	Inhibit cell growth & cell proliferation Induce apoptosis via DNA degradation Induce cell death via mitochondrial depolarization	0–100 μM	N/A	[52]
	A431	Inhibit cell viability & proliferation Induce G0/G1 phase cell cycle arrest: ↓ cyclin A, cyclin D1, cyclin D2 & cyclin E; ↓ Cdk2, Cdk4 & Cdk6; ↑ p21 & p27 Induce cell apoptosis: ↑ PARP	0–75 μM	N/A	[53]
	B16-F10	Inhibit cell proliferation Induce cell death: ↑ Autophagosome (vacuoles) formation; ↓ cyclin D1; ↓ AKT/mTOR & Notch signalling	0–50 μM	N/A	[54]
	B16/F-10 & SKMEL-28	Inhibit cell proliferation & viability: ↓ Notch signalling; ↓ TACE & γ-secretase complex proteins Inhibit clonogenicity Induce G0/G1 phase cell cycle arrest Induce autophagy: ↑ autophagosome formation; ↑ LC3B cleavage Inhibit cell stemness: ↓ CD271, CD166, Jarid1B & ABCB5	0–60 μM	N/A	[55]
	UACC903	Inhibit cell growth & proliferation	0–50 μM	7.45 μM (24 h) 5.10 μM (72 h)	[24]
	SKMEL-2	Inhibit cell proliferation & viability Induce apoptotic death: ↑ caspase-3, caspase-6, caspase-8 & caspase-9; ↑ PARP cleavage; ↓ procaspase-3, procaspase-8 & procaspase-9 Induce G2/M phase cell cycle arrest: ↓ cyclin B1, cyclin D1, cyclin D2 & PCNA; ↓ Cdk2 & Cdk4; ↑ p21 & p53	0–100 μM	N/A	[56]
	UACC-62	Inhibit cell proliferation & viability Induce apoptotic death: ↑ caspase-3, caspase-6, caspase-8 & caspase-9; ↑ cleaved PARP; ↓ procaspase-3, procaspase-8 & procaspase-9 Induce G0/G1 phase cell cycle arrest: ↓ cyclin B1, cyclin D1 & cyclin D2; ↓ Cdk2, Cdk4 & Cdc2p34; ↓ p21 & p27	0–100 μM	N/A	[56]
Renal cancer	A498	Inhibit cell proliferation Inhibit colony formation capability Inhibit cell migration and invasion: ↓ Epithelial-mesenchymal transition (EMT); ↓ cancer stem cells (CSC) properties; ↑ miR-141; ↓ ZEB2 Inhibit tumoursphere formation	0–80 μM	~12 μM (72 h)	[57]
Cervix cancer	KB-3-1, KB-8-5, KB-C1, KB-V1	Inhibit cell viability: ↓ EGFR-STAT3 signalling Induce mitochondria-dependent & death receptor-dependent apoptosis: ↓ Bcl-2, Mcl-1 & survivin; ↑ PARP & caspase-3 cleavage; ↑ mitochondrial release of cytochrome c; ↑ DR5 Enhances in vitro cytotoxicity of Paclitaxel	0–75 μM	KB-3-1: 12.56 μM (72 h) KB-8-5: 12.08 μM (72 h) KB-C1: 11.40 μM (72 h) KB-V1: 10.39 μM (72 h)	[31]
Pancreatic cancer	MiaPaCa & Colo-357	Suppress plating efficiency of cells Reduce anchorage-independent clonogenicity growth Suppress migration and invasion ability	0–5 μM	N/A	[58]
	MiaPaCa & Panc1	Inhibit cell growth Induce G1 phase cell cycle arrest: ↓ cyclin D1 & cyclin E; ↓ Cdk2 & Cdk4; ↑ p21 & p27 Induce apoptosis: ↓ Bcl-2 & Bcl-xL proteins; ↑ Bax protein; ↓ IKB-α phosphorylation; ↓ NF-κB constitutive activation	0–60 μM	MiaPaCa: 43.25 μM (24 h) 31.08 μM (48 h) 18.54 μM (72 h) Panc1: 47.44 μM (24 h) 34.17 μM (48 h) 21.86 μM (72 h)	[59]
Thyroid cancer	ARO, WRO	Inhibit cell growth & proliferation: ↓ ERK, JNK & p37 activation and expression; ↓ mTOR & p70S6K Inhibit colony formation Induce apoptosis: ↑ PARP cleavage; ↑ caspase-3, caspase-8 & PARP activation; ↓ PI3K/AKT & MAPK pathways Induce G0/G1 cell cycle arrest: ↓ cyclin D1; ↓ Cdk2 & Cdk4; ↑ p21 & p27 Induce autophagy & autophagy flux: ↑ LC3-II	ARO & WRO: 0–60 μM SW579: 0–40 μM	ARO: 36.3 μM (24 h) 40.1 μM (48 h) 44.8 μM (72 h) WRO: 37.7 μM (24 h) 31.8 μM (48 h) 30.7 μM (72 h) SW579: 19.9 μM (24 h) 10.5 μM (48 h) 8.8 μM (72 h)	[60]
Nasopharyngeal cancer	HNE-1	Inhibit cell growth Induce apoptosis Induce G1 phase cell cycle arrest	0–150 μM (Honokiol & ATNH—Active targeting nanoparticles-loaded honokiol)	Honokiol: 144.71 μM (24 h) ATNH: 69.04 μM (24 h)	[30]
Brain cancer	U251	Inhibit cell growth Inhibit cell proliferation Induce apoptosis	0–120 μM	61.43 μM (24 h)	[61]
	T98G	Inhibit cell viability Inhibit cell invasion Induce cell apoptosis: ↑ Bax protein; ↓ Bcl-2; ↑ Bax/Bcl-2 ratio	0–50 μM	N/A	[62]
	GBM8401 (Parental) & GBM8401 SP	Inhibit cell proliferation & viability Induce sub-G1 phase cell cycle arrest Induce apoptosis: ↓ Notch3/Hes1 pathway	0–20 μM	GBM8401 (Parental): 5.30 μM (48 h) GBM8401 SP: 11.20 μM (48 h)	[36]
	U251 & U-87 MG	Inhibit cell viability & proliferation: ↓ PI3K/Akt & MAPK/Erk signalling pathways Inhibit cell invasion & migration: ↓ MMP2 & MMP9; ↓ NF-κB-mediated E-cadherin pathway Inhibit colony formation Induce apoptosis: ↓ Bcl-2, p-AKT & p-ERK; ↑ Bax protein; ↑ caspase-3 cleavage; ↓ EGFR-STAT3 signalling Reduce spheroid formation: ↓ CD133 & Nestin protein	0–60 μM	U251: 54.00 μM (24 h) U-87 MG: 62.50 μM (24 h)	[63]
	DBTRG-05MG	Inhibit cell growth Induce apoptosis: ↓ Rb protein; ↑ PARP & Bcl-x(S/L) cleavage Induce autophagy: ↑ Beclin-1 & LC3-II	0–50 μM	~30 μM	[64]
	U87 MG (Human) BMEC (Mouse)	Inhibit cell viability Inhibit epithelial-mesenchymal transition (EMT): ↓ Snail, β-catenin & N-cadherin; ↑ E-cadherin Inhibit cell adhesion & invasion: ↓ VCAM-1; ↓ phosphor-VE-cadherin-mediated BMEC permeability	0–20 μM	U87MG: 22.66 μM (24 h) BMEC: 13.09 μM (24 h)	[65]
	U87 MG	Inhibit cell viability Induce G1 phase cell cycle arrest: ↑ p21 & p53; ↓ cyclin D1; ↓ Cdk4 & Cdk6; ↓ p-Rb protein; ↓ E2F1 Induce apoptosis: ↓ procaspase-3; ↑ caspase-8 & caspase-9 activity	0–100 μM	52.70 μM	[66]
Bone cancer	HOS & U20S	Inhibit cell proliferation Inhibit colony formation Induce G0/G1 phase cell cycle arrest: ↓ cyclin D1 & cyclin E; ↓ Cdk4 Induce mitochondria-mediated apoptosis: ↑ caspase-3 & caspase-9 activation; ↑ PARP cleavage; ↓ Bcl-2, Bcl-xL & survivin; ↑ ERK activation; ↓ proteasome activity; ↑ ER stress and subsequent ROS overgeneration; ↑ GRP78 Induce autophagy: ↑ Atg7 protein activation; ↑ Atg5; ↑ LC3B-II	0–30 μM	HOS: 17.70 μM (24 h) U20S: 21.50 μM (24 h)	[67]
	SAOS-2, HOS, 143B, MG-63 M8, HU09, HU09 M132 Dunn, LM5, LM8 & LM8-LacZ (Mouse)	Inhibit cell metabolic activity Inhibit cell proliferation Inhibit cell migration Induce rapid cell death via Honokiol-provoked vacuolation	0–150 μM	(72 h) SAOS-2: 48.38 μM HOS: 51.38 μM 143B: 41.63 μM MG-63M8: 34.88 μM HU09: 59.25 μM HU09M132: 31.88 μM (72 h) Dunn: 36.00 μM LM5: 30.00 μM LM8: 31.13 μM	[68]
	Saos-2 & MG-63	Inhibit cell viability Induce apoptosis: ↑ caspase-3 & PARP cleavage; ↑ Bax protein; ↓ Bcl-2; ↓ PI3K/AKT signalling pathway; ↓ miR-21	0–100 μM	Saos-2: 37.85 μM (24 h) MG-63: 38.24 μM (24h)	[69]
Oral cancer	OC2 & OCSL	Inhibit cell growth Induce G0/G1 phase cell cycle arrest: ↑ cyclin E accumulation; ↑ p21 & p27; ↓ cyclin D1, ↓ Cdk2 & Cdk4 Induce apoptosis: ↓ caspase-8 & caspase-9; ↑ caspase-3 cleavage; ↓ Bid protein Induce autophagy and autophagic flux: ↑ LC3-II; ↓ Akt/mTORC1 pathway; ↑ AMPK signalling pathway; ↑ p62	0–60 μM	OC2: 35.00 μM (24 h) 22.00 μM (48 h) OCSL: 33 μM (24 h) 13 μM (48 h)	[26]
	HN-22 & HSC-4	Inhibit cell viability Induce apoptosis: ↓ Sp1 protein; ↑ p21 & p27; ↑ PARP & caspase-3 activation; ↓ Mcl-1 & survivin protein Induce G1 phase cell cycle arrest: ↓ cyclin D1	0–37.5 μM	HN-22: 26.63 μM (48 h) HSC-4: 30.00 μM (48 h)	[70]
Liver cancer	HepG2	Inhibit cell growth & proliferation: ↓ β-catenin protein Induce apoptosis: ↑ BAD protein; ↓ Bcl-2 protein Upregulation of BAD protein expression Downregulation of Bcl-2 protein level	0–2 μM	N/A	[71]
	SMMC-7721	Inhibit cell growth Induce G0/G1 phase cell cycle arrest Induce apoptosis: ↓ mitochondrial potential; ↑ ROS production; ↓ Bcl-2 protein; ↑ Bax protein	0–37.5 μM	N/A	[72]
	HepG2, HUH7, PLC/PRF5, Hep3B	Inhibit cell proliferation: ↓ STAT3 activation; ↓ IL-induced Akt phosphorylation; ↓ c-Src activation; ↓ JAK1 & JAK2; ↑ SHP-1 protein Induce sub-G1 phase cell cycle arrest: ↓ cyclin D1 Downregulation of cyclin D1 level Induce apoptosis: ↓ Bcl-2 & Bcl-xL; ↓ survivin & Mcl-1 protein; ↑ caspase-3 activation; ↑ PARP cleavage Enhance apoptotic effect of doxorubicin & paclitaxel	0–100 μM	N/A	[32]
Ovarian cancer	A2780s & A2780cp	Inhibit cell growth Induce apoptosis	0–100 μM	A2780s: 36.00 μM (48 h) A2780cp: 34.70 μM (48 h)	[73]
	SKOV3 & Caov-3	Inhibit cell proliferation and growth Inhibit colony formation Induce apoptosis: ↑ AMPK pathway activation; ↑ caspase-3, caspase-7 & caspase-9 activation; ↑ PARP cleavage Induce G0/G1 phase cell cycle arrest Inhibit cell migration and invasion	0–100 μM	SKOV: 48.71 μM (24 h) Caov-3: 46.42 μM (24 h)	[28]
	SKOV3, COC1, Angelen & A2780	Inhibit cell proliferation Induce cell apoptosis: ↓ Bcl-xL; ↑ BAD protein; ↑ caspase-3 activation Induce G1 phase cell cycle arrest	0–93.75 μM	SKOV3: 62.63 μM (24 h) COC1: 73.50 μM (24 h) Angelen: 61.50 μM (24 h) A2780: 55.85 μM (24 h)	[74]
Prostate cancer	PC-3 & LNCaP	Inhibit cell viability Induce G0/G1 phase cell cycle arrest: ↓ cyclin D1 & cyclin E; ↓ Cdk2, Cdk4 & Cdk6; ↑ p21 & p53; ↓ Rb & E2F1 proteins; ↓ Rb phosphorylation at Ser^807/811^; ↑ ROS generation	0–60 μM	N/A	[75]
	PC-3, LNCaP & C4-2	Inhibit cell growth Induce apoptosis: ↑ caspase-3, caspase-8 & caspase-9 activation; ↑ PARP cleavage Induce apoptosis via DNA fragmentation: ↑ Bax & Bak proteins; ↓ Mcl-1 protein	0–75 μM	18.75–37.50 μM (24 h)	[76]
	PC-3, LNCaP	Inhibit cell viability Induce autophagy: ↑ LC3-BII protein; ↓ mTOR pathway Induce apoptosis via DNA fragmentation: ↑ ROS generation	0–40 μM	N/A	[77]
Head & neck squamous cancer	Cal-33 & MD-1483	Inhibit cell growth Induce cell apoptosis and cell cycle arrest: ↓ EGFR signalling pathway; ↓ STAT3 signalling pathway; ↓ Bcl-xL & cyclin D1; ↓ phosphorylation p42/p44 MAPK & phosphorylated Akt	0–100 μM	Cal-33: 3.80 μM (72 h) 1483: 7.44 μM (72 h)	[78]
Neuroblastoma	Neuro-2a	Induce apoptosis via DNA fragmentation: ↑ caspase-3, caspase-6 & caspase-9 activation; ↑ Bax protein; ↓ mitochondrial membrane potential; ↑ cytochrome c releaseInduce sub-G1 phase cell cycle arrest	0–100 μM	63.3 μM (72 h)	[79]
	Neuro-2a & NB41A3	Inhibit cell viability Induce autophagy: ↑ LC3-II; ↑ PI3K/Akt/mTOR signalling pathway; ↑ Grp78; ↑ ROS generation; ↑ ERK1/2; ↑ p-ERK1Induce apoptosis via DNA fragmentation Inhibit cell migration	0–100 μM	Neuro-2a: ~50 μM (72 h)	[80]
Bladder cancer	T24 & 5637	Inhibit cell viability and induce apoptosis: ↑ Bax protein; ↑ PARP cleavage; ↓ Bcl-2 protein Inhibit clonogenicity Induce G1 phase cell cycle arrest: ↓ cyclin D1; ↑ p21 & p27 Inhibit sphere formation capacity Inhibit cell migration & invasion: ↓ EZH2 gene expression; ↓ MMP9 Inhibit cell stemness: ↓ EZH2 gene expression; ↓ CD44 & Sox2; ↑ miR-143 overexpression	0–72 μM	N/A	[81]

### 4.2. In Vivo Studies

Based on the in vivo studies, honokiol possessed the capability to inhibit tumour growth, metastasis, and angiogenesis using different animal models, as shown in Table 2. The degree of tumour inhibition was shown to be significantly effective against each distinct cancer cell line, ranging from 0–150 mg/kg via various delivery methods of honokiol between oral gavage or injection (intraperitoneal, caudal vein, or intravenous). Honokiol was shown to downregulate the expression of Oct4, Nanog, and Sox2 which were known to be expressed in osteosarcoma, breast carcinoma and germ cell tumours [41]. According to Wang et al.’s study, they have found that the average tumour size was significantly lower than the control group without affecting their body weight, suggesting inconsequential toxicity under tested conditions when treated with a combination of honokiol and paclitaxel [31]. Indisputably, honokiol was once again proven to exhibit minor to no toxicity against normal cells.

Over the years, the development of chemo-resistance in ovarian cancer cells has hindered the outcome of treatment regimen towards ovarian cancer [82]. Despite the effectiveness of honokiol to inhibit cancer cell proliferation, delivering effective concentration towards the tumour site was deemed challenging due to its water insolubility [73]. The encapsulation of honokiol in liposome, namely Lipo-HNK by Luo and his team has displayed substantial efficacy against cisplatin-resistance ovarian cancer cell line A2780cp. The tumour volume for Lipo-HNK treated mice was 408 ± 165 mm^3^ compared to liposome-treated mice and control mice were 2575 ± 701 mm^3^ and 2828 ± 796 mm^3^ respectively after 21 days [73]. In addition, Lipo-HNK was also shown to prolong survival and induce intra-tumoral apoptosis in vivo. The promising in vivo properties of honokiol should consolidate its importance as a potential anticancer agent for future researches.

Zebrafish (*Danio rerio*) model has emerged as a newly important cancer model that complements against traditional cell culture assays and mice model due to its small size, heavy brood, and rapid maturation time. Importantly, its transparent body wall enables visibility of tumour progression and the ease of experimentation [83,84]. It was known that juvenile zebrafish (*Danio rerio*) or zebrafish embryos have the competency to study cancer cell invasion, metastasis, tumour-induced angiogenesis. Honokiol reduced U-87 MG human glioma/glioblastoma cell proliferation and migration in zebrafish yolk sac and in vivo xenograft nude mouse model [63]. These observations are associated with a reduction in EGFR, phosphorylated STAT3, CD133 and Nestin levels, thus highlighting the regulation of honokiol in EGFR-mediated STAT3/JAK signalling pathway to induce anti-tumour and anti-metastasis.

The subsections below will further discuss the mechanism of anticancer actions of honokiol including the induction of cancer cell death, inhibition of cell cycle progression, induction of autophagy, prevention of epithelial–mesenchymal transition (EMT), as well as the suppression of migration, invasion, and angiogenesis of cancer cells.

**Table 2 cancers-12-00048-t002:** The antitumour effect of honokiol in in vivo tumour bearing animal models.

Cancer Cell Line	Animal Model & Site of Tumour Xenograft	Dose, Duration & Route of Administration	Observation & Mechanism of Action	Efficacy on Tumour Inhibition	References
Breast cancer					
MDA-MB-231 cells	Both flanks of athymic nude mice	100 mg/kg/day 28 days IP	Induce tumour growth arrest	Complete arrest of tumour growth from week 2 onwards	[39]
MDA-MB-231 cells	Right gluteal region of athymic nude mice	3 mg/mouse/day Three times a week 28 days IP	Inhibit tumour progression: ↓ Ki-67; ↑ LKB1 & pAMPK; ↑ ACC phosphorylation, ↓ pS6K & 4EBP1 phosphorylation	Tumour weight of honokiol-treated group was 0.22 g compared to control group which was 1.58 g	[40]
MDA-MB-231-pLKO.1 & MDA-MB-231-LKB1^shRNA^ cells	Right gluteal region of athymic nude mice	3 mg/mouse/day Three times a week 42 days Oral gavage	Inhibit cell stemness: ↓ Oct4, Nanog & Sox2; ↓ pSTAT3 & Ki-67 Inhibit mammosphere formation	Decreased expression of Oct4, Nanog, Sox2 Reduce number of tumour cells showing Ki-67 & pStat3 expression	[41]
Colorectal cancer					
RKO cells	Axilla of BALB/c nude mice	80 mg/kg/day Treatment on days 8–11, 14–17, 21–24, 28–31 51 days IP	Inhibit tumour growth Prolong survival of mice	709.9% increase in tumour growth rate in honokiol-treated group compared to 1627.6% and 1408.2% in control and vehicle groups respectively	[33]
HCT116 cells	Flank of athymic nude mice	200 μg/kg/day + 5 Gy irradiation Once a week 21 days IP	Inhibit tumour growth: ↓ CSC proteins → ↓ DCLK1, Sox-9, CD133 & CD44	Significantly lower tumour weight (<800 mg) in honokiol-IR combination, (~1500 mg) in honokiol treatment group compared to (~3300 mg) in control group	[35]
Cervical cancer					
KB-8-5 cells	Athymic nu/nu nude mice (site of xenograft not stated)	50 mg/kg Honokiol Three times a week + 20 mg/kg Paclitaxel Once a week 28 days IP (honokiol) Tail vein injection (paclitaxel)	Suppress tumour growth: ↓ Ki-67 tissue level Induce apoptosis	Significantly lower average tumour volume for honokiol-paclitaxel combination treatment (573.9 mm^3^) compared to control (2585.4 mm^3^)	[31]
Lung cancer					
H2030-BrM3 cells	Left ventricle of NOD/SCID mice	2 or 10 mg/kg/day 28 days Oral gavage	Prevent metastasis of lung cancer cells to brain	10 mg/kg: Decrease brain metastasis for >70%	[47]
H2030-BrM3 cells	Left lung via left ribcage of athymic nude mice	2 or 10 mg/kg/day Five days a week 28 days Oral gavage	Decrease lung tumour growth Inhibit metastasis to lymph node	10 mg/kg: Significantly reduce incidence of mediastinal adenopathy, decrement of weight of mediastinal lymph node for >80%, only 2/6 mice have lymphatic metastasis	[47]
Blood cancer					
Raji cells	Back of BALB/c nude mice	5 mg/20 g & 10 mg/20 g Treatment on days 8–12 & 15–19 20 days (Route of administration not specified)	Inhibit cell proliferation Inhibit tumour growth	Tumour growth of honokiol-treated mice was significantly lower (~90 cm^3^) compared to control mice (~270 cm^3^)	[38]
HL60 cells	Inoculated intraperitoneally into SCID mice	100 mg/kg/day Treatment on Day 1–6 47 days IP	Prolong survival of mice	Median survival time of honokiol-treated mice are longer (37.5 days) compared to vehicle-treated mice (24.5 days)	[85]
Pancreatic cancer					
MiaPaCa cells	Pancreas of immunocompromised mice	150 mg/kg/day 28 days IP	Suppress tumour growth Inhibit metastasis: ↓ CXCR & SHH; ↓ NF-κB & downstream pathway Inhibit desmoplastic reaction: ↓ ECM protein; ↓ collagen I	Significant decrease in tumour growth for honokiol-treated mice (99.6 mm^3^) compared to vehicle-treated mice (1361.0 mm^3^)	[58]
Skin cancer					
SKMEL-2 or UACC-62 cells	Right flank of athymic nude mice	50 mg/kg Three times a week 14–54 days IP	Decrease tumour growth	SKMEL-2: 40% reduction in tumour volume UACC-62: 50% reduction in tumour volume	[56]
Thyroid cancer					
ARO cells	BALB/cAnN.Cg-Foxn1nu/CrlNarl mice (site of xenograft not stated)	5 or 15 mg/kg/mouse Every three days 21 days Oral gavage	Decrease tumour volume & tumour weight Induce apoptosis & autophagy	Control: ~1000 mm^3^; 700 mg 5 mg/kg Honokiol: ~600 mm^3^; 400 mg 15 mg/kg Honokiol: ~400 mm^3^; 200 mg	[60]
Nasopharyngeal cancer					
HNE-1 cells	Right dorsal aspect of right foot of BALB/c athymic nude mice	Active-targeting nanoparticles-loaded HK (ATNH), Non-active-targeting nanoparticles-loaded HK (NATNH), Free Honokiol (HK) 3 mg/mouse/day Every three days Euthanise 50% mice after 12 days, rest are left to observe tumour growth & survival time up to 60 days; IV	Inhibit tumour progression, Induce apoptosis Potential inhibitor of angiogenesis & proliferation	Efficiency in tumour delay: ATNH > NATNH > Free HK Median survival time: Control: 28.5 days Free HK: 34 days NATNH: 42.5 days ATNH: 57.5 days	[30]
Brain cancer					
U21 cells	Right flank of athymic nude mice	20 mg/kg Twice a week 27 days Caudal vein injection	Inhibit tumour growth Inhibit angiogenesis	Honokiol-treated mice have significant inhibition of tumour volume by 50.21% compared to vehicle Significantly lower microvessel present in honokiol-treated cells	[61]
U-87 MG cell suspension pre-treated with honokiol or vehicle for 48h	Yolk sac of Zebrafish larvae	(Concentration N/A) 3 days Injection of cells into zebrafish	Inhibit cell proliferation Inhibit cell migration	Reduced number of cell mass compared to vehicle-treated cells	[63]
U-87 MG cells	Right flank near upper extremity of nude mice	100 mg/kg/day Treatment at days 1–7 21 days IP	Reduce tumour growth: ↓ EGFR, pSTAT3, CD133 & Nestin	Increased number of apoptotic cells in honokiol-treated tissue, Significantly lower tumour volume & tumour weight in honokiol-treated mice	[63]
Bone cancer					
HOS cells	Dorsal area of BALB/c-nu mice	40 mg/kg/day 7 days IP	Reduce tumour growth Induce apoptosis & autophagy: ↑ cleaved caspase-3; ↑ LC3B-II & phosphor-ERK (ROS/ERK1/2 signalling pathway)	Significant decrease in tumour volume & weight of honokiol-treated mice (200 mm^3^; 0.2 g) compared to control group (~500 mm^3^; 0.5 g) Increased number of TUNEL-positive cells	[26]
LM8-LacZ cells	Left flank of C3H/HeNCrl mice	150 mg/kg/day 25 days; IP	Inhibit metastasis	Mean number of micrometastases decreased significantly by 41.4% in honokiol-treated mice compared to control mice	[68]
Oral cancer					
SAS cells	Right flank of BALB/cAnN.Cg-Foxn1nu.CrlNarl nude mice	5 mg/kg or 15 mg/kg Treatment on day 1, 4, 7, 10, 13, 16, 19, 22 35 days Oral	Reduce tumour growth & volume	Significantly reduction in tumour growth in honokiol-treated mice 29% reduction (5 mg/kg; 21 days), 40% reduction (15 mg/kg; 21 days) 41% reduction (5 mg/kg; 35 days), 56% reduction (15 mg/kg; 35 days)	[26]
Prostate cancer					
C4-2 cells	Bilateral tibia of BALB/c nu/nu athymic nude mice	100 mg/kg/day 42 days IP	Inhibit cell proliferation: ↑ Ki-67 Induce apoptosis: ↑ M-31 Inhibit angiogenesis: ↑ CD-31	Lower PSA value in honokiol-treated mice compared to control group	[76]
PC-3 cells	Left & right flanks above hind limb of nude mice	1 or 2 mg/mice Monday, Wednesday & Friday two weeks before tumour implantation and duration of experiment after implantation 77 days Oral gavage	Inhibit tumour growth Inhibit cell proliferation Inhibit neovascularisation Induce apoptosis	Tumour volume of honokiol-treated mice are significantly lower (~330 mm^3^; 1 mg), (~50 mm^3^; 2 mg) compared to control (~400 mm^3^)	[18]
Gastric cancer					
MKN45 cells	Dorsal side of BALB/c nude mice (nu/nu)	0.5 mg/kg/day & 1.5 mg/kg/day 10 days Injection (route not stated)	Inhibit tumour growth: ↓ GRP94 overexpression	30% reduction in tumour volume (0.5 mg/kg) 60% reduction in tumour volume (1.5 mg/kg) Decreased accumulation of GRP94	[86]
MKN45 & SCM-1 cells	Peritoneal cavity of BALB/c nude mice	5 mg/kg Twice a week 28 days IP	Inhibit metastasis Inhibit angiogenesis	Honokiol inhibited STAT-3 signalling and VEGF signalling induced by calpain/SHP-1	[87]
Ovarian cancer					
SKOV3 cells	Right axilla of BALB/c nude mice	1 mg liposome-encapsulated honokiol/day 48 days IP	Inhibit tumour growth Inhibit angiogenesis	Reduction in tumour growth rate in liposome-encapsulated honokiol-treated mice by 67–70% compared to control	[73,88]
A2780s cells	Right flank of athymic BALB/c nude mice	10 mg/kg Lipo-Honokiol Twice a week 21 days IV	Inhibit cancer growth Prolong survival of mice Increase intra-tumoural apoptosis Inhibit intra-tumoural angiogenesis	Lipo-HNK treated mice have significantly smaller tumour volume (222 ± 71 mm^3^) compared to liposome-treated mice (1823 ± 606 mm^3^) and control mice (3921 ± 235 mm^3^)	[73]
A2780cp cells	Right flank of athymic BALB/c nude mice	10 mg/kg Lipo-Honokiol Twice a week 21 days IV	Inhibit cancer growth Prolong survival Increase intra-tumoural apoptosis Inhibit intra-tumoural angiogenesis	Lipo-HNK treated mice have significantly smaller tumour volume (408 ± 165 mm^3^) compared to liposome-treated mice (2575 ± 701 mm^3^) and control mice (2828 ± 796 mm^3^)	[73]

## 5. Mechanism of Action of Honokiol

### 5.1. Dual Induction of Apoptotic and Necrotic Cell Death

Apoptosis is a normal physiological process that maintains the homeostatic cellular balance in multicellular organisms [89]. Generally, apoptosis can be classified into two central pathways, namely the intrinsic pathway (mitochondrial-mediated pathway) and extrinsic pathway (death receptor-mediated pathway) [90]. The intrinsic pathway is associated with changes in mitochondrial membrane permeability that lead to imbalance in Bax/Bak ratio and release of cytochrome *c* and other mitochondrial proteins into cytosol [89,90]. The released cytochrome *c* interacts with apoptosis protease-activating factor 1 (Apaf1) and forms an apoptosome complex [91], which promotes the activation of caspase-9 and later caspase-3, initiating the caspase cascade, which executes cell death in a coordinated way [91]. For the extrinsic pathway, the binding of ligands such as tumour necrosis factor (TNF), Fas ligand (Fas-L), and TNF-related apoptosis-inducing ligand (TRAIL) to their respective death receptors (type 1 TNF receptor (TNFR1), Fas (also called CD95/Apo-1) and TRAIL receptors will turn procaspase-8 into active caspase-8 to induce apoptosis [91,92,93].

Honokiol has been shown tp initiate caspase-dependent apoptotic pathways in different types of cancer (Table 1). Chen et al. [14] found that JJ012 human chondrosarcoma cells lose their mitochondrial membrane potential when treated at 10 µM of honokiol, thus leading to apoptosis. Other studies have also shown that honokiol markedly disrupted the balance of Bax/Bcl-2 ratio [13,18,34,63,94,95,96,97]. The increasing ratio of proapoptotic to antiapoptotic Bcl-2 family proteins (Bax/Bcl-2) will induce the release of cytochrome *c* and other apoptogenic proteins through the mitochondrial membrane to the cytosol, subsequently leading to the activation of caspase cascade and apoptosis [34]. Furthermore, honokiol downregulated the expression of several other anti-apoptosis mRNA and proteins such as Bcl-xL [13,18,25,64], survivin [67,98], and MCL-1 [18], as well as upregulated other pro-apoptotic proteins such as BAD, BAX, and BAK proteins [18,25].

Moreover, honokiol has been shown to effectively induce apoptosis in p53-deficient cancer cells, such as MDA-MD-231 breast cancer cells, as well as lung and bladder cancer cell lines by inhibiting the activation of ras-phospholipase D [39,99,100]. Besides p53, other tumour suppressor genes that will be activated in honokiol treatment include p21 [53], p21/waf1 [101], p27 [53], p38 MAPK [102,103], and p62 [26,46].

Besides the intrinsic pathway, honokiol is capable of targeting death receptors TNF-related apoptosis-inducing ligand (TRAIL) receptors and tumour necrosis factor receptors (TNFR) resulting in a sequential activation of caspase-8 and -3, which cleaves target proteins and then leads to apoptosis [104,105,106]. Activation of the death receptor mediated apoptotic pathway is primarily inhibited by cellular-caspase-8/FADD-like IL-1β-converting enzyme (FLICE) inhibitory protein (c-FLIP), which inhibits caspase-8 activation by preventing the recruitment of caspase-8 to the death inducing signalling complex [106]. However, honokiol was able to downregulate c-FLIP through the ubiquitin/proteasome-mediated mechanism, resulting in the sensitisation of non-small cell lung cancer cells to TRAIL-mediated apoptosis [107,108].

Other than intrinsic and extrinsic pathways, honokiol can also induce apoptosis by the endoplasmic reticulum (ER) stress-induced mechanism. A variety of ER stresses result in unfolded protein accumulation responses [109,110]. For survival, the cells induce ER chaperone proteins to increase protein aggregation, temporarily halt translation, and activate the proteasome machinery to degrade misfolded proteins. However, under severe and prolonged ER stress, an unfolded protein response activates unique pathways that lead to cell death through apoptosis [111]. According to a study by Zhu et al. [50], honokiol can upregulate the expressions of ER stress-induced apoptotic signalling molecules such as GRP78, phosphorylated PERK, phosphorylated eIF2α, CHOP, Bcl-2, Bax, and cleaved caspase-9 in human lung cancer cells. Chiu et al. [112] found that honokiol also led to an increase in ER stress activity in melanoma cell lines B16F10 (mouse), human malignant melanoma, and human metastatic melanoma. Honokiol activated ER stress and down-regulated peroxisome proliferator-activated receptor-γ (PPARγ) activity resulting in PPARγ and CRT degradation through calpain-II activity in human gastric cancer cell lines [86,113,114] and human chondrosarcoma cells [14]. This was due to the ability of honokiol to upregulate and bind effectively to the glucose regulated protein 78 (GRP78) to activate apoptosis [14,115]. However, this was opposed by another study where treatment of various human gastric cancer cells with honokiol led to the induction of GRP94 cleavage but did not affect GRP78 [86].

Necrosis is known as unprogrammed cell death whereby cell swelling and destabilisation of the cell membrane results in the leakage of cellular cytoplasmic contents into the extracellular space, thus causing inflammation [116]. Besides apoptosis, honokiol has also been found to induce necrotic cell death in MCF-7 (40 μg/mL honokiol) [117], human oesophageal adenocarcinoma cells CP-A and CP-C [118], and primary human acute myelogenous leukemia HL60 [85] via p16ink4a pathway by targeting cyclophilin D to affect several downstream mechanisms. This phenomenon was also observed in transformed Barrett’s and oesophageal adenocarcinoma cells when treated with honokiol (<40 µM) by targeting their STAT3 signalling pathway, thus resulting in a decrease of Ras activity and phosphorylated ERK1/2 expression [119]. The phosphorylation of Ser727 STAT3 induces translocation towards the mitochondria followed by ROS production, ultimately leading to the induction of necrosis [120]. Taken together, honokiol demonstrates the dual induction of apoptotic and necrotic cell death.

### 5.2. Cell Cycle Arrest

Cancer is attributed to uncontrolled proliferation resulting from abnormal activity of different cell cycle proteins. Therefore, cell cycle regulators are becoming attractive targets in cancer therapy. Honokiol can induce cell cycle arrest in several types of cancer cells, such as in lung squamous cell carcinoma [121], prostate cancer cells [75,122], oral squamous cancer [70], UVB-induced skin cancer [123], and more as listed in Table 1, by generally inducing G0/G1 and G2/M arrest. This arrest is associated with the suppression of cyclin-B1, CDC2, and cdc25C in honokiol-treated human gastric carcinoma and human neuroglioma cells [97,124,125], downregulation of cyclin dependent kinase (CDK)-2 and CDK-4, and the upregulation of cell cycle suppressors p21 and p27 in human oral squamous cell carcinoma (OSCC) cells [26,97]. In addition, the downregulation of c-Myc and class I histone deacetylases was also identified as other contributors to cell cycle arrest at the G0/G1 phase in prostate cancer cells [97,122] and acute myeloid leukemia respectively [44,101,108].

### 5.3. Autophagy

Autophagy is an evolutionary conserved catabolic process that involves the delivery of dysfunctional cytoplasmic components for lysosomal degradation [126,127]. The activation of autophagy promotes cell survival and regulates cell growth during harsh and stressful conditions via a reduction of cellular energy requirements by breaking down unnecessary components [82,127]. In cancer cells, autophagy facilitates both tumour suppression and tumourigenesis by the induction of cell death and tumour growth promotion, respectively [128,129]. The regulation of mTORC complexes mTORC1 and mTORC2 is involved in controlling the autophagic process. The activation of mTORC1 plays an important role in phosphorylation of autophagy-related protein (ATG) and subsequently inhibiting autophagy, whereas the inhibition of mTORC1 complements the autophagic process [130,131]. The inhibition of mTORC1 complex will concurrently activate Unc-51-like autophagy-activating kinase (ULK) complex, inducing localisation to the phagophore and followed by class III PI3K activation [132,133]. Beclin-1 was known to play a role in tumour suppression by recruiting several proteins associated with autophagosome elongation and maturation [134]. ATGs regulate the autophagosome elongation. For instance, ATG5-ATG12/ATG16L complexes recruit microtubule-associated protein 1 light chain 3 (LC3), followed by conversion of pro-LC3 to active cytosolic isoform LC3 I by ATG4B [135,136]. Thereafter, the interaction with ATG3, ATG7, and phosphatidylethanolamine (PE) converts LC3 I to LC3 II. The LC3 II enables the autophagosome to bind to degraded substrates and mature autophagosomes are capable of fusing with lysosomes to selectively remove damaged organelles via autophagy [137].

Generally, there are two modes of autophagy known as conventional and alternative autophagy. Conventional autophagy (also known as Atg5/Atg7-dependent pathway) involves the activation of Atg5 and Atg7 which are core regulators of autophagy, and then leads to microtubule-associated protein 1A/1B light chain 3 (LC3) modification and translocation from cytosol to the isolation membrane. This LC3 translocation was considered as a reliable hallmark of autophagy. Contradictorily, alternative autophagy occurs independently without involving Atg5 and Atg7, as well as LC3 modification [128,129,137].

The regulation of autophagy in cancer remains controversial as it plays dual roles in tumour suppression and promotion. Autophagy is believed to contribute to the properties of cancer cells stemness, induction of recurrence, and the development of anticancer drugs. However, the actual mechanism of autophagy in cancer remains unclear. Several studies have highlighted the potential of honokiol to induce cell death via autophagy in human prostate cancer cells [77], human glioma cells [138], NSCLC cells [30], and human thyroid cancer cells [60].

The activation of Atg5/Atg7-dependent pathways through the upregulation of LC3B-II, Atg5, and Atg7 levels was observed in honokiol-treated osteosarcoma HOS and U2OS cells and leads to the accumulation of autophagic vacuoles [26]. According to a study by Chang et al. [64], the expression of two critical autophagic proteins, Beclin-1 and LC3, were found to have increased in the honokiol-treated glioblastoma multiforme cells (DBTRG-05MG cell line). Similarly, the expression of autophagosomal marker LC3-II was also increased in Kirsten rat sarcoma viral oncogene homolog (KRAS) mutated cell lines of non-small cell lung cancer (NSCLC).

Other signalling pathways are also found to be involved in honokiol-induced autophagy including the involvement of AMPK-mTOR signalling pathway which leads to autophagocytosis through the coordinated phosphorylation of Ulk1 in Kirsten rat sarcoma viral oncogene homolog (KRAS) mutant lung cancer and melanoma cells [55,60,66,97]. Besides this, the ROS/ERK1/2 signalling pathway is also believed to play a certain role in honokiol-induced autophagy though ERK activation and the generation of ROS in treated osteosarcoma cells [67,77,97]. All these recent studies have further supported the potential of honokiol in the induction of autophagy in cancer cells.

### 5.4. Epithelial-Mesenchymal Transition (EMT)

Migratory mesenchymal-like cells are involved in embryonic development, tissue repair, and regeneration, as well as several pathological processes like tissue fibrosis, tumour invasiveness, and metastasis [139,140]. These migratory mesenchymal cells originate from the conversion of the epithelial cells, and this process is known as epithelial-mesenchymal transition (EMT). This plasticity of cellular phenotypes provides a new insight into possible therapeutic interventions in cancer [140].

EMT is characterised by the loss of epithelial markers such as cytokeratins and E-cadherin, followed by an increase in mesenchymal markers such as N-cadherin and vimentin [141]. The cellular processes of EMT are composed of several key transcription factors (such as TWIST, SNAI1, SNAI2, ZEB1/2) that act in concert with epigenetic mechanisms and post-translational protein modifications to coordinate cellular alterations [139,142]. The application of gene expression signatures combining multiple EMT-linked genes has proven useful to evaluate EMT as a contributing factor in tumour development in human cancers. However, the EMT process was shown to be incomplete in tumours, venturing in between multiple translational states and expressing a mixture of both epithelial and mesenchymal genes. This hybrid in partial EMT can be more aggressive than tumour cells with a complete EMT phenotype [141]. In addition, EMT contributes to tumour metastatic progression and resistance towards cancer treatment, resulting in poor clinical outcomes [140,141].

Honokiol has been shown to block and inhibit EMT in many cancer cells such as breast cancer, melanoma, bladder cancer, human non-small cell lung cancer, and gastric cancer (Table 1). Honokiol reduced steroid receptor coactivator-3 (SRC-3), matrix metalloproteinase (MMP)-2, and Twist1, preventing the invasion of urinary bladder cancer cells [108,143]. In addition, honokiol was also capable of inducing E-cadherin and repressing N-cadherin expression, thus inhibiting the EMT process in J82 bladder cancer cells [108,143]. In breast cancer cells, honokiol inhibits the recruitment of Stat3 on mesenchymal transcription factor Zeb1 promoter, resulting in decreased Zeb1 expression and nuclear translocation [144]. In addition, honokiol increases E-cadherin expression via the Stat3-mediated release of Zeb1 from E-cadherin promoter [144]. Collectively, many studies have reported that honokiol effectively inhibits EMT in breast cancer cells, evidence has been found to support a cross-talk between honokiol and Stat3/Zeb1/E-cadherin axis [144]. On the other hand, EMT is inhibited by modulating the miR-141/ZEB2 signalling in renal cell carcinoma (A-498) [57].

Honokiol inhibited the EMT-driven migration of human NSCLC cells in vitro by targeting c-FLIP through N-cadherin/snail signalling as N-cadherin and snail are downstream targets of c-FLIP [145]. Twist1, a basic helix-loop-helix domain-containing transcription factor, promotes tumour metastasis by inducing EMT, and can be upregulated by multiple factors, including SRC-1, STAT3, MSX2, HIF-1α, integrin-linked kinase, and NF-κB. The capability of honokiol in targeting Twist1 can be regarded as a promising therapy for metastatic cancer [108,146].

Honokiol was found to inhibit breast cancer cell metastasis and eliminate human oral squamous cell carcinoma cell by blocking EMT through the modulation of Snail/Slug protein translation [147,148]. Honokiol markedly downregulated endogenous Snail, Slug, and vimentin expression and upregulated E-cadherin expression in MDA-MB-231, MCF7, and 4T1 breast cancer cells [148]. As primary EMT inducers, Snail and Slug dictate the induction of EMT by targeting E-cadherin and vimentin [144,148]. Furthermore, when cells were treated with honokiol, Snail and Slug expression levels were decreased from 12 h to 24 h in a time-dependent manner, suggesting that honokiol can reverse the EMT process via the downregulation of Snail and Slug in breast cancer cell lines [148]. Besides that, EMT was inhibited in human oral squamous cell carcinoma cell via the disruption of Wnt/β-catenin signalling pathway [147]. It was reported that the protein levels of mesenchymal markers such as Slug and Snail were markedly suppressed, while β-catenin and its downstream Cyclin D1 were inhibited [147]. It is known that β-catenin could mediate EMT [147,149], which plays a crucial role in cancer invasion and metastasis. The EMT markers such as Snail and Slug are also the target genes of β-catenin [150]. Therefore, the suppression of Snail and Slug in honokiol treated human oral squamous cell carcinoma cells was believed to be due to the inhibition of Wnt/β-catenin signalling pathway [147]. Similarly, in U87MG human glioblastoma cell and melanoma cells, Snail, N-cadherin and β-catenin expression levels were decreased, whereas E-cadherin expression was increased after honokiol treatment [65,112].

### 5.5. Suppression of Migration, Invasion and Angiogenesis of Cancer Cells

Metastasis is known to be the major cause of death in cancer patients [151]. It involves the migration and invasion of tumour cells into neighbouring tissues and distant organs via intravasation into blood or lymphatic system [152,153]. The formation of invadopodium was stimulated by epidermal growth factor (EGF) and is crucial for the degradation of the extracellular matrix and remodelling membrane proteins, promoting metastasis [151]. Therefore, one of the important steps in cancer management is to control tumour cell metastasis, especially for early-stage cancer patients [153]. Various studies have reported that honokiol has the capability to suppress tumour metastasis in different types of cancer including breast cancer [40,148,154], non-small cell lung cancer [44,155] ovarian carcinoma cells [28], lung cancer [50], U251 human glioma, as well as U-87MG and T98G human glioblastoma cell [63,65,94], oral squamous cell carcinoma (OSCC) [26], bladder cancer cell [143], pancreatic cancer [58], renal cell carcinoma [156,157], and gastric cancer cells [113]. For instance, the percentage of invading urinary bladder cancer (UBC) cells was significantly reduced by 67% and 92% upon 2.4 μg/mL and 4.8 μg/mL of honokiol treatment, respectively [143]. Similarly, tumour cell migration was inhibited by 38–66% in A549 cells, by 37–62% in H1299 cells, 12% to 58% in H460 cells and 32% to 69% in H226 cells, in a concentration-dependent manner after treatment with honokiol [44].

Furthermore, honokiol also demonstrated an inhibitory effect on the expression of matrix metalloproteinases (MMPs) such as MMP-2 and MMP-9 proteins, which play an essential role in the metastatic process of tumour cells, as well as the regulation of angiogenesis in the maintenance of tumour cell survivability [44,63,143]. MMPs are a group of extracellular matrix degrading enzymes that control various normal cellular processes, such as cell growth, differentiation, apoptosis, and migration [153]. However, MMP activity was increased in many tumour cells. The overexpression of MMP-2 and MMP-9 are associated with pro-oncogenic events such as neovascularisation, tumour cell proliferation, and metastasis because it can degrade the extracellular matrix, basement membranes, and adhesion molecules (intercellular adhesion molecule, ICAM, and vascular cell adhesion molecule) and become invasive [58,153,158].

The transition from an epithelial-to-mesenchymal (EMT) phenotype facilitates the breakdown of extracellular matrix followed by the subsequent invasion of the surrounding tissues in order to enter the bloodstream and/or lymph nodes, and travel to distant organ sites. Once cells have reached the distant organ sites, they undergo mesenchymal-to-epithelial transition and begin the establishment of distal metastasis by the surviving cancer cells followed by the outgrowth of secondary tumours [58,159]. Honokiol has been shown to inhibit the invasion of HT-1080 human fibrosarcoma cells and U937 leukemic cells by inhibiting MMP-9 [160]. In addition, honokiol also reduced the protein levels of MMP2 and MMP9 in U251 human glioma and U-87 MG human glioblastoma cell lines in a dose-dependent manner [63]. The expression of MMP-2 and MMP-9 were also found to be decreased in both honokiol-treated A549 and H1299 cells (NSCLC cell lines), consistent with the decreased nuclear accumulation of β-catenin as both MMP-2 and MMP-9 are the downstream targets of β-catenin [44,161,162]. In the J82 bladder cancer cell, honokiol repressed the expression of SRC-3, MMP-2, and Twist1 genes which were involved in cancer cell invasion [143].

Another proposed mechanism for the inhibitory effects of honokiol on invasion and metastasis is through the liver kinase B1 (LKB1)/adenine monophosphate-activated protein kinase (AMPK) axis. Honokiol treatment increased the expression and cytoplasmic translocation of tumour-suppressor LKB1 in breast cancer cells, which led to the phosphorylation and functional activation of AMPK and resulted in the inhibition of cell invasion and metastasis [40,58]. The activation of AMPK suppresses mTOR signalling, decreasing the phosphorylation of p70 kDA ribosomal protein S6 kinase 1 (p70S6K1) and eukaryotic translation initiation factor 4E (eIF4E)-binding protein (4EBP1). This will ultimately inhibit the reorganisation of the actin cytoskeleton in cells, subsequently inhibiting cell migration [40].

In human renal carcinoma cell (RCC) 786-0, honokiol significantly upregulated the expression of metastasis suppressor gene (KISS-1), genes encoding TIMP metalloproteinase inhibitor 4 (TIMP4), and KISS-1 receptor (KISS-1R). In addition, honokiol markedly suppressed the expression of genes encoding chemokine (C-X-C motif) ligand 12 (CXCL12), chemokine (C-C motif) ligand 7 (CCL7), interleukin-18 (IL18) and matrix metalloproteinase 7 (MMP7). It was proven that honokiol significantly upregulated KISS1 and KISS1R in the 786-0 cells when treated with honokiol since recent studies showed that the activation of KISS1/KISS1R signalling by kisspeptin treatment decreases the motility and invasive capacity of conventional RCC, and overexpression of KISS1 inhibits the invasion of RCC cells Caki-1 [14,163]. In short, the activation of KISS1/KISS1R signalling by honokiol suppresses the multistep process of metastasis, including invasion and colony formation, in RCC cells 786-0 [163].

Angiogenesis is the formation of new blood vessels for supplying nutrients and oxygen to tissues and cells. In tumourigenesis, angiogenesis is important for the development and progression of malignant tumours [164]. The endothelial cells in growing cancer are active due to the release of cell growth and motility promoting proteins, creating a network of blood vessels to overcome its oxygen tension [165]. Vascular endothelial growth factor (VEGF) and fibroblast growth factor-2 (FGF2) are among the factors that play an important role in tumour angiogenesis [153]. In human renal cancer cell lines (786-0 and Caki-1), honokiol induced down-regulation of the expression of VEGF and heme oxygenase-1 (HO-1) via the Ras signalling pathway thus inhibit angiogenesis [166,167].

In retinal pigment epithelial (RPE) cell lines, honokiol inhibited the binding of hypoxia- inducible-factor (HIF) to hypoxia-response elements present on the VEGF promoter, thereby inhibiting the secretion of VEGF protein [168,169]. This decrement of VEGF levels resulted in reduced proliferation of human retinal microvascular endothelial cells (hRMVECs) [168]. Therefore, honokiol is said to possess both anti-HIF and anti-angiogenic properties.

In the overexpression of VEGF-D Lewis lung carcinoma cell-induced tumours in C57BL/6 mice, honokiol was shown to significantly inhibit tumour-associated lymphangiogenesis and metastasis. Furthermore, a remarkable delay in tumour growth and prolonged life span in honokiol-treated mice were also observed [170]. In another study, honokiol inhibited VEGF-D-induced survival, proliferation, and microcapillary tube formation in both human umbilical vein endothelial cells (HUVECs) and lymphatic vascular endothelial cells (HLECs). These observations are believed to be due to the inhibition in Akt and MAPK phosphorylation and downregulation of VEGFR-2 expressions in HUVECs as well as VEGFR-3 of HLECs [101,160,171]. Collectively, honokiol has been shown to exert direct and indirect effects on tumour suppression via anti-metastasis, anti-angiogenesis, and anti-lymphangiogenesis by mainly affecting HIF- and VEGF/VEGFR- dependent pathways. However, an in-depth mechanism of honokiol on the inhibition of metastatic progression and spread should be further explored in the future.

## 6. Effect of Honokiol on Various Signalling Pathways

### 6.1. Nuclear Factor Kappa B (NF-κB)

The nuclear factor kappa B (NFκB) family comprises of five DNA-binding proteins (p50, p52, p65, cRel, and RelB) that differentially modulate the transcription of genes that are involved in various cellular processes such as inflammation, migration, invasion, angiogenesis, proliferation, and apoptosis [172,173]. The continuous activation of NFκB has been reported in different types of cancers. Honokiol affects the constitutive activation of NFκB and expression of NFκB-regulated gene products involved in apoptosis (survivin, Bcl-2, Bcl-xL, IAP1, IAP2, cFLIP and TRAF1), inflammation (cyclooxygenase-2, COX-2), proliferation (cyclin D1 and c-myc), invasion (ICAM-1 and MMP-9), and angiogenesis (VEGF), thereby enhancing apoptosis and suppressing cancer progression [58,174]. Several studies support the inhibitory activity of honokiol against NFκB in different types of cancer cells, including breast cancer [42,117,175], head and neck squamous cell carcinoma (HNSCC) [176], colon cancer cells [177], non-small cell lung cancer (NSCLC) cells [44], pancreatic cancer cells [13], human leukemic cell [104], embryonic kidney cells, T-cell leukemia, multiple myeloma, lung adenocarcinoma, and squamous cell carcinoma [174].

Honokiol was found to repress the transcriptional activity of NFκB in both pancreatic MiaPaCa and Panc1 cancer cells. It was found that honokiol treatment significantly reduced nuclear NFκB levels with an increase of cytoplasmic NFκB fraction in MiaPaCa and Panc1 cells, in a dose-dependent manner [13]. The cellular distribution of NFκB is controlled by the relative expression of its biological inhibitor IκB, which keeps NFκB sequestered in the cytoplasm in an inactive complex [172]. Upon honokiol treatment, IκB-α levels were increased due to the stabilisation of IκB-α post-treatment, concurrently inducing the downregulation of IκB-α phosphorylation [13]. Furthermore, honokiol has also been shown to inhibit the TNF-α-induced phosphorylation and degradation of the cytosolic NFκB inhibitor IκBa and suppression of IKK activation [104,174,178]. In addition, honokiol was also found to inhibit the nuclear translocation and phosphorylation of p65 subunit of NFκB [44,104]. Honokiol suppressed NF-κB-regulated gene products including MMP-9, TNF-α, IL-8, ICAM-1, and MCP-1 [66].

### 6.2. Signal Transducers and Activators of Transcription (STATs)

Signal transducers and activators of transcription (STATs) is a well-known oncogene that is regulated by receptor tyrosine kinases, G-protein-coupled receptors, and interleukin families [179,180]. STAT3 are a group of transcription factors that upon phosphorylation will undergo dimerization and translocation to either the nucleus or mitochondria to control cell survival, cell cycle, cellular growth, and angiogenesis. STATs are aberrantly activated in several types of malignancies due to functional loss of their negative regulators, or the overexpression of upstream tyrosine kinases [179]. STAT3 can also localise into the mitochondria and mediate mitochondrial biogenesis. Honokiol has been shown to target STAT3 to reduce its expression and phosphorylation in many cancer cells such as human glioblastoma [47,63,100], lung cancer [47,181], oral squamous cell carcinoma (OSCC) [95], breast cancer [41,144], human epidermoid carcinoma [31], colorectal cancers [182], gastric cancer [87], and esophageal adenocarcinoma [119].

Honokiol was found to inhibit EGFR expression and down-regulate STAT3 phosphorylation by reducing the CD133 and Nestin levels [63]. Similarly, honokiol also induces apoptosis through the suppression of JAK2/STAT3, Akt and Erk signalling pathways in human oral squamous cell carcinoma (SAS and OCEM-1) cell lines [95]. Similar effect was observed in oral cancer cells where honokiol suppressed JAK2/STAT3 activation and, inhibited IL-6-mediated cell migration [95,183]. Furthermore, another study indicated that honokiol induces apoptosis in human glioblastoma cell line U87 through suppressing the phosphorylation of STAT3 (Tyr705), down-regulating survivin, and upregulating cleaved caspase-3 expression [98].

Moreover, honokiol inhibited STAT3-phosphorylation/activation in an LKB1-dependent manner, preventing its recruitment to canonical binding-sites in the promoters of Nanog, Oct4, and Sox2 [41]. Thus, the inhibition of the coactivation function of STAT3 resulted in the suppression of expression of pluripotency factors in MCF7, MDA-MB-231, SUM149, and SUM159 breast cancer cells [41]. Furthermore, honokiol inhibited breast tumorigenesis in mice in an LKB1-dependent manner [41]. This showed that honokiol can support crosstalk between LKB1, STAT3, and pluripotency factors in breast cancer and effective anticancer modulation of this axis with honokiol treatment in both in vitro and in vivo [41]. Apart from that, honokiol suppressed metastasis and proliferation in both brain metastatic lung cancer cell lines PC9-BrM3 and H2030- BrM3 by inhibiting STAT3 phosphorylation [47].

In other studies, honokiol is proven to be an effective chemotherapeutic agent that exert its antitumour function by inhibiting the STAT3 signalling pathway. Honokiol can induce cell cycle arrest and apoptosis via the inhibition of survival signals in adult T-cell leukemia by suppressing the phosphorylation and DNA binding of different oncogene factors, such as NF-κB, activator protein 1, STAT3, and STAT5 [184]. Besides that, honokiol can induce necrosis and apoptosis in transformed Barrett’s and oesophageal adenocarcinoma cells through the inhibition of the STAT3 signalling pathway [119]. Honokiol can inhibit the growth and peritoneal metastasis of gastric cancer in nude mice, which was correlated with the inhibition of STAT3 signalling via the upregulation of Src homology 2 (SH2)-containing tyrosine phosphatase 1 [87].

### 6.3. Epidermal Growth Factor Receptor (EGFR)

EGFR is a group of transmembrane receptor tyrosine kinases (RTKs) that are normally deregulated in various cancers [185,186]. The overexpression or activating mutations in EGFR results in increased cell proliferation, abnormal metabolism, and cell survival through the activation of the downstream mitogen-activated protein kinase (MAPK) and v-akt murine thymoma viral oncogene homolog 1 (AKT) signalling pathways, as well as phosphatidyl-inositol 3-kinase (PI3K)/Akt, and STAT3 signalling pathways [13,58]. EGFR activation occurs upon binding to its ligands, which then leads to its homo- or heterodimerization with other members of the ErbB family, and subsequent activation of downstream signalling cascades in many cancer cell types, including breast cancer and head and neck squamous cell carcinoma (HNSCC) [187,188].

Honokiol has been shown to inhibit EGFR signalling pathway through either inhibition of EGFR expression or inhibition of EGFR phosphorylation [78,189,190]. Honokiol (60 µM) was found to inhibit EGFR expression and down-regulate STAT3 phosphorylation in U251 and U-87 MG human glioma/glioblastoma cells via JAK-STAT3 signalling [63]. In another study, honokiol (2.5–7.5 μM) differentially suppressed proliferation (up to 93%) and induced the apoptosis (up to 61%) of EGFR overexpressing tumourigenic bronchial cells. These effects were observed in parallel with the downregulation of phospho-EGFR, phospho-Akt, phospho- STAT3, and cell cycle-related proteins [189]. Furthermore, in a mouse lung tumour bioassay, intranasal instillation of liposomal honokiol (5 mg/kg) for 14 weeks reduced the size and multiplicity (49%) of lung tumours and the level of total- and phospho-EGFR, phospho-Akt, and phospho-STAT3 [189]. Overall, honokiol has been proven to be a promising candidate to suppress the development and progression of lung tumours driven by EGFR deregulation. Moreover, honokiol induced mitochondria-dependent and death receptor-mediated apoptosis in multi-drug resistant (MDR) KB cells, which was associated with inhibition of EGFR-STAT3 signalling and downregulation of STAT3 target genes [31].

Furthermore, the downregulation of c-Src/EGFR-mediated signaling is involved in honokiol-induced cell cycle arrest and apoptosis in MDA-MB-231 human breast cancer cells. EGFR can also be activated in a ligand-independent manner by cellular Src (c-Src), a non-receptor tyrosine kinase. The tyrosine kinase c-Src is also upregulated in many human malignancies and promotes the activation of mitogenic signalling through EGFR [13,191]. In MDA-MB-231 human breast cancer cells, honokiol downregulated the expression and phosphorylation of c-Src, epidermal growth factor receptor (EGFR), and Akt, and consequently led to the inactivation of mTOR and its downstream signal molecules including 4E-binding protein (4E-BP) and p70 S6 kinase [43]. Besides that, inhibition of HER-2 signalling by specific human epidermal growth receptor 1/HER-2 (EGFR/HER-2) kinase inhibitor lapatinib synergistically enhanced the anti-cancer effects of honokiol in HER-2 over-expressed breast cancer cells [42].

The treatment of HNSCC cells with honokiol also decreased the expression of total EGFR as well as p-EGFR and its downstream target, mTOR. Since the activation of mTOR has been shown to contribute to tumour progression, it can be speculated that the honokiol-induced inhibition of cell proliferation in HNSCC cells is mediated through the downregulation of EGFR/mTOR signalling pathway [176,192]. These observations are consistent with the evidence that honokiol inhibits the growth of cancer cells by targeting EGFR and its downstream molecular targets and suggest that these mechanisms are in play in HNSCC.

### 6.4. Mammalian Target of Rapamycin (mTOR)

The mammalian target of rapamycin (mTOR) is a type of protein kinase which regulates cell metabolism, proliferation, and growth. The activation of PI3K/Akt pathway results in the aberrant activation of mTOR in most cancer cells [97,193,194]. It is known that mTOR controls the expression of many survival proteins via activating p70 S6 kinase (S6K) and inhibition of eIF4E inhibitor 4E-BP1 [193]. The mTOR signalling pathway is dysregulated in premalignant or early malignant human tissues and is highly implicated in the carcinogenic process. Honokiol suppresses the activation of mTOR and its signalling mediators (4E-BP1 and p70 S6 kinase) by inhibiting ERK and Akt pathways [43] or upregulating PTEN (Phosphatase and Tensin homolog) expression [42,157].

Honokiol was found to induce apoptosis and suppress migration and invasion in ovarian carcinoma cells (SKOV3 and Caov-3) via TSC1/TSC2 complex/AMPK/mTOR signalling pathway [28]. This is mediated via the regulation of the tumour suppressors p27, p53, and MMP-9 [28]. Furthermore, it was proven that honokiol was able to attenuate PI3K/Akt/mTOR signalling via the down-regulation of Akt phosphorylation and upregulation of PTEN expression in breast cancer cells (MCF-7, MCF-7/adr, and BT-474 cell lines) [42]. A combination of honokiol with the mTOR inhibitor rapamycin presented synergistic effects to induce apoptosis in breast cancer cells where the inhibition of PI3K/Akt/mTOR signalling by the mTOR inhibitor further sensitizes breast cancer cells to honokiol [42]. Other studies have also shown that honokiol induces autophagy in PC-3 and LNCaP prostate cancer cells via the suppression of mTOR and Akt phosphorylation [77]. Another study revealed that the treatment of neuroblastoma cells with honokiol caused significant downregulation of mTOR phosphorylation, which leads to the induction of autophagy of neuroblastoma cells (neuro-2a cells) through the PI3K/Akt/mTOR signalling pathways [96,195].

### 6.5. Hypoxia-Inducible-Factor (HIF) Pathway

The master regulator of neovascularisation, HIF, is a transcription factor that that plays an integral role in the body’s response to low oxygen concentrations (i.e., hypoxia) [196,197]. Active HIF is composed of of two subunits: HIF-α and HIF-1/ARNT. Transcriptional regulation by oxygen is mediated by the HIF-α isoforms. In humans, three isoforms of α-subunit (HIF-1α, HIF-2α, and HIF-3α) have been identified. Recent studies suggest that transcriptional adaptation to hypoxia involves epigenetic changes in histone methylation. Strong evidence has established that the expression of pro-angiogenic factors (VEGF), which play a critical role in pathological neovascularisation in cancer, is elevated due to the activation of HIF pathway under hypoxia conditions [198].

An activation of the HIF pathway leading to hypoxia-induced neovascularisation is the central cause of pathogenesis in almost all solid tumours and ischemic retinal diseases [198,199]. There are studies reporting the capability of honokiol to inhibit HIF isoforms and the expression of hypoxic markers, as well as the binding of HIF to hypoxia-response elements present on VEGF promoter in D407 cells (human retinal pigment epithelial cells) [168]. In KRAS mutant lung cancer cells, it was discovered that Sirt3 was significantly up-regulated in honokiol-treated KRAS mutant lung cancer cells, leading to the destabilisation of its target gene Hif-1α and induction of G1 arrest and apoptosis. This suggests that the anticancer property of honokiol is regulated via a novel mechanism associated with the Sirt3/Hif-1α [49].

### 6.6. Notch Signalling Pathway

Notch signalling has been implicated in maintaining tissue homeostasis, including the regulation of self-renewal in adult stem cells, organ development, and embryonic development [200,201,202]. In mammals, the Notch receptor family comprises of four receptors (Notch-1, Notch-2, Notch-3, and Notch-4) and five ligands (Delta-like-1, Delta-like-3, Delta-like-4, Jagged-1, and Jagged-2). Each Notch receptor is activated through cell membrane-associated ligands. A series of proteolytic cleavage processes lead to the maturation and activation of Notch receptors. The first cleavage was catalysed by ADAM-family metalloprotease TACE, followed by the second cleavage mediated by γ-secretase, an enzyme complex that contains presenilin, nicastrin, presenilin enhancer 2 (PEN2), and anterior pharynx-defective 1 (APH1). The series of cleavages will lead to the release and translocation of Notch intracellular domain (NICD) into the nucleus [202]. Activated NICD is able to bind to activator proteins, including mastermind-like proteins (MAML) and recombination signalling binding protein-J (RBPJ) to form a nuclear transcriptional activator complex to regulate the transcription of downstream target genes, such as the hairy and enhancer of split (Hes) gene, Hey family genes, c-myc, cyclin D1, and p21/Waf1 [200]. The Notch pathway plays a complex role in the tumourigenesis of both hematologic and solid tissues. In fact, Notch signalling plays a vital role in regulating cellular differentiation, angiogenesis, proliferation, and apoptosis [201].

It has been shown that honokiol can eliminate cancer stem-like cells and potentiation of temozolomide (TMZ) sensitivity in glioblastoma multiforme (GBM) cells [36]. It was shown that honokiol enhanced the sensitization of GBM cells to MGMT inhibitor O6 benzylguanine (O6-BG) through the downregulation of Notch3 as well as the expression of its downstream target, Hes1 [36]. Furthermore, honokiol has been shown to inhibit B16/F-10, SKMEL-28 melanoma cell lines and SW480 colon cancer cells by targeting Notch signalling pathways [203,204]. Honokiol treatment resulted in reduced levels of cleaved Notch, particularly the Notch-2 receptor, along with a decrease in the expression of downstream target proteins, including Hes-1, cyclin D1, as well as TACE and γ-secretase complex proteins in melanoma cells [55].

Apart from that, honokiol in combination with radiation treatment reduced the number of DCLK1+ (cancer stem cell marker protein) colon cancer cells, which was accompanied by reduced levels of activated Notch-1, its ligand Jagged-1, and the downstream target gene Hes-1 [35,204]. Furthermore, the expression of components of the Notch-1 activating γ-secretase complex, presenilin 1, nicastrin, Pen2, and APH-1 were also suppressed [35]. To determine the effect of a honokiol–IR combination on tumour growth in vivo, nude mice tumour xenografts were administered honokiol intraperitoneally and exposed to IR. The honokiol–IR combination significantly inhibited tumour xenograft growth [35]. In addition, there were reduced levels of DCLK1 and the Notch signalling–related proteins in the xenograft tissues. Together, these data suggest that honokiol is a potent inhibitor of colon cancer growth that targets the stem cells by inhibiting the γ-secretase complex and the Notch signalling pathway [35,204].

### 6.7. Downregulation of P-Glycoprotein

The principal mechanism of multidrug resistance is due to the active transport of drugs out of cells [205]. Among the efflux transporters, P-glycoprotein (P-gp, gene symbol ABCB1) plays an important role in the resistance of cancer cells to a variety of chemotherapeutic treatments [205,206]. Furthermore, P-gp is distributed throughout the body where it interacts with various drugs of different structures to limit their bioavailability [207]. Therefore, the development of effective inhibitors of P-gp expression and/or functional activity should reverse drug resistance and enhance the bioavailability of P-gp substrates. One of the effective ways to overcome P-gp mediated drug resistance is either to block its drug-pump function or to inhibit its expression. To date, there are a total of three generations of P-gp inhibitors that have been discovered [207,208]. However, these compounds were not used widely due to toxicity at the doses required for attenuating P-gp activity, poor specificity, or unpredictable pharmacokinetic interactions. Honokiol was shown to downregulate the expression of P-gp at mRNA and protein levels in MCF-7/ADR, a human breast MDR cancer cell line [209,210]. The downregulation of P-gp was accompanied by a partial recovery of intracellular drug accumulation [210]. In MDR ovarian cancer cells (NCI/ADR-RES), honokiol has also been shown to downregulate the expression of P-gp in a concentration- and time-dependent manner [208].

## 7. Metabolism, Bioavailability, and Pharmacological Relevance of Honokiol

Pharmacokinetics involves the study of drug movement within the body, which includes the time course of absorption, distribution, metabolism, and excretion (ADME). Honokiol is mainly metabolized in the liver and undergoes in vivo biotransformation, whereby glucuronidation and sulfation are the main metabolic pathways to convert honokiol into mono-glucuronide honokiol and sulphated mono-hydroxyhonokiol before elimination [23]. This extensive biotransformation of honokiol may contribute to its low bioavailability. Currently, studies are being conducted to determine whether the metabolites of honokiol possess any biological activities that can extend the half-life of honokiol while maintaining its biological properties.

Most of the studies have reported that honokiol undergoes a rapid distribution and absorption, but slow elimination after intravenous (i.v.) administration [13,58,211,212]. For i.v. administration, it has been found that there was a rapid rate of distribution followed by a slower rate of elimination (elimination half-life t_1/2_ = 49.22 min and 56.2 min for 5 mg or 10 mg of honokiol, respectively) observed in Sprague Dawley rats [213]. In another study, Liang et al. [214] investigated the pharmacokinetic properties of honokiol in beagle dogs after intravenous guttae, whereby the blood plasma of both male and female dogs was assessed. The elimination half-life (t_1/2_ in hours) was found to be 20.13 (female), 9.27 (female), 7.06 (male), 4.70 (male), and 1.89 (male) after administration of doses of 8.8, 19.8, 3.9, 44.4, and 66.7 mg/kg, respectively. The t_1/2_ decreases with an increase in the dose and length of infusion [214]. In another study, Wang et al. [61] discovered for the first time that honokiol is able to cross the blood–brain barrier (BBB) and blood–cerebrospinal fluid barrier (BCSFB) after i.v. administration when tested on intracerebral gliosarcoma model in Fisher 344 rats and human U251 xenograft glioma model in nude mice. It was also reported that the honokiol was distributed in the order of: lungs > plasma > liver > brain > kidney > heart > spleen after i.v. administration [61].

Furthermore, honokiol has been studied via an intraperitoneal route of administration. Chen et al. [33] reported a maximum plasma concentration of honokiol at 27.179 ± 6.252 min, with the t_1/2_ of 312.08 ± 51.66 min after intraperitoneal injection of 250 mg/kg in BALB/c mice. On another note, studies have also shown that the presence of rhubarb and immature orange fruit extract in the decoction influenced the pharmacokinetics of honokiol, where a single oral dose of honokiol in Houpu decoction (a compound prescription of honokiol; 5 g/kg body weight) in Wistar rats demonstrated an elimination t_1/2_ of 526.6 min [215]. Honokiol has a rapid absorption (Tmax = 20 min) and slow elimination (t_1/2_z = 290 min) after a single dose of oral gavage at 40 mg/kg in healthy rats [216]. In another study, honokiol showed a peak plasma concentration at 72 min, and t_1/2_ of 186 min, and the absolute bioavailability for honokiol was found to be 5.3% when rats underwent oral administration of Magnolol/Honokiol emulsion (4:1) at 50 mg/kg [217]. After the rats were administered with honokiol orally, the honokiol was distributed rapidly to all parts of organs with the highest concentration being accumulated in the liver, followed by the brain and kidneys [216]. This was opposed to their discovery in tumour-bearing mice, where the highest concentration was found in the liver, followed by the kidneys and lungs [218]. This may be due to the different types of species being used as well as the tumor-burdened mice possibly affecting drug distribution [47]. With the rectal administration of Houpo extract at a dose of 245 mg/kg (equivalent to 13.5 mg/kg of honokiol) in Wistar rats, the maximal plasma concentration of honokiol found was approximately six times to that administered orally at an identical dose, indicating that rectal dosing avoids first-pass metabolism to some extent [219].

Meanwhile, the topical application of honokiol on UVB-induced contact hypersensitivity (CHS) as a model in C3H/HeN mice was also evaluated [68,220]. The topical application of honokiol (0.5 and 1.0 mg/cm^2^ skin area) had a significant preventive effect on the UVB-induced suppression of the CHS response. The inflammatory mediators COX-2 and PGE_2_ played a key role in this effect, as indicated by the honokiol-mediated inhibition of cyclooxygenase-2 (COX-2) expression and PGE_2_ production in the UVB-exposed skin. Besides that, both topical application and oral administration of honokiol significantly inhibited (38% to 46%, *p* < 0.001) UVB-induced suppression of CHS in mice compared with the mice that were not treated with honokiol but exposed to UVB radiation. Prominently, the level of inhibition of CHS was not significantly different between the two modes of administration of honokiol [220].

Apart from that, Gao et al. [221] investigated the enhancement in the transdermal and localised delivery of honokiol through breast tissue. It was reported that microneedle-porated dermatome significantly increased the delivery of honokiol by nearly three-fold (97.81 ± 18.96 μg/cm^2^) compared with passive delivery (32.56 ± 5.67 μg/cm^2^). Oleic acid was found to be the best chemical penetration enhancer, increasing the delivery almost 27-fold (868.06 ± 100.91 μg/cm^2^). The addition of oleic acid also resulted in a better retention of drugs in porcine mammary papilla (965.41 ± 80.26 μg/cm^2^) compared with breast skin (294.16 ± 8.49 μg/cm^2^) [221]. In summary, both microneedles and chemical enhancers can improve the absorption of honokiol through the skin. Directly applying honokiol on mammary papilla is a potential administration route which can increase localized delivery into breast tissue [183].

On another note, some studies have addressed the poor solubility of honokiol in hydrophilic environment. Wang et al. [222] developed polyethylene glycol-coated (PEGylated) liposomal honokiol to improve its solubility compared to free honokiol. PEGylated (polyethylene glycol coated) liposomal honokiol was shown to enhance the serum honokiol concentration and decrease clearance. The pharmacokinetic analysis of PEGylated liposomal honokiol showed a two-fold increase in elimination t_1/2_ value as compared to that of free honokiol when being injected through the i.v. route (20 mg/kg body weight) in Balb/c mice (from 26 min in PEGylated liposomal honokiol to 13 min in free honokiol) [222]. Moreover, the AUC_0→∞_ (mean concentration of drug in plasma) of PEGylated liposomal honokiol was about 1.85-fold higher than free honokiol. The protein-binding ability of honokiol in plasma was reported to be between 60% and 65% as revealed by equilibrium dialysis [222]. In another study, plasma honokiol concentrations were maintained above 30 and 10 μg/mL for 24 and 48 h, respectively, in liposomal honokiol-treated mice. However, it was reduced rapidly (<5 μg/mL) by 12 h in free honokiol-treated mice bearing A549 xenograft tumors, suggesting that liposomal honokiol extended blood circulation times in tumor-bearing mice compared to free honokiol [223].

## 8. Potential Drug Delivery of Honokiol

Due to the low water solubility and bioavailability of honokiol, multiple studies have been performed to develop proper honokiol delivery systems to improve its pharmacological effectiveness. A few studies have been performed to develop efficient drug carriers to deliver honokiol to its respective target, including the development of nanoparticles [224,225,226], micelles [227,228,229], and liposomes [73,171,223].

For honokiol delivery in the form of nanoparticles, Zheng et al. [230] developed monomethoxy poly(ethylene glycol)–poly(lactic acid) (MPEG–PLA) via ring opening polymerisation and then processed into nanoparticle for honokiol delivery. The honokiol-loaded MPEG–PLA nanoparticles were mono-dispersed and stable in the aqueous solution [230]. It was found that only 53% of honokiol was released from the nanoparticles within 24 h, while 100% of free honokiol was released into the outside media, suggesting that the honokiol loaded MPEG–PLA nanoparticle is a novel honokiol formulation which could meet the requirement of intravenous injection. In comparison, honokiol loaded MPEG-PLA nanoparticles significantly decreased the viability of A2780s cells (human ovarian cancer cells) than free honokiol, indicating that honokiol loaded MPEG–PLA nanoparticles might possess great potential applications for anticancer effect on cisplatin-sensitive A2780s cells in vitro [230]. In addition, the incorporation of both honokiol and doxorubicin in MPEG-PLA nanoparticles exhibited stronger anticancer activity than its individual form against A2780s cells [231].

In another study, emulsion solvent evaporation was used to develop the active targeting nanoparticle-loaded honokiol (ATNH) using copolymerpoly (ε-caprolactone)-poly (ethylene glycol)-poly (ε-caprolactone) (PCEC), which was modified with folate (FA) by introducing polyethylenimine (PEI) [30]. It was reported that ATNH showed a suitable size distribution, high encapsulation efficiency, gradual release, and targeting uptake by human nasopharynx carcinoma cells (HNE-1). Moreover, ATNH significantly inhibited tumour growth, metabolism, proliferation, micro-vessel generation, and caused cell-cycle arrest at the G1 phase [30]. Apart from that, epigallocatechin-3-gallate functionalized chitin loaded with honokiol nanoparticles (CE-HK NP), developed by Tang et al. [224], inhibit HepG2 cell growth and induce apoptosis through the suppression of mitochondrial membrane potential. Furthermore, CE-HK NPs (40 mg/kg) inhibited tumour growth by 83.55% (*p* < 0.05), which was far higher than the 30.15% inhibition of free honokiol (40 mg/kg). The proposed delivery system exhibits better tumour selectivity and growth reduction in both in vitro and in vivo models (male BALB/c nude mice treated with honokiol administrated by intertumoral injection) and did not induce any side effects [224]. Therefore, the CE-HK NPs may act as an effective delivery system for liver cancer. Recently, Yu et al. [232] further improved the design of nanoparticles for targeted delivery in breast cancer by surface modifying the honokiol nanoparticles through conjugation with folic acid to the surface of honokiol nanoparticles coated with polydopamine (HK-PDA-FA-NPs) as a pH-sensitive targeting anchor for nanoparticles. The targeted nanoparticles (HK-PDA-FA-NPs) can be stably present in various physiological media and exhibit pH sensitivity during drug release in vitro. HK-PDA-FA-NPs have better targeting ability to 4T1 cells than normal HK-NPs. Targeted nanoparticles have a tumour inhibition rate of greater than 80% in vivo (female Balb/c mice injected intraperitoneally with 40 mg/kg HK-PDA-FA-NPs), which is significantly higher than conventional HK-NPs [232].

For honokiol delivery in the form of micelles, researchers developed poly(ethylene glycol)-poly(ε-caprolactone)-poly(ethylene glycol) (PECE) micelle loaded with honokiol [229]. The cytotoxicity results showed that the composite drug delivery system is a safe carrier and the encapsulated honokiol retained its potent antitumor effect when tested against murine melanoma cell line B16 [233]. The IC_50_ values of free honokiol, honokiol nanoparticles, and honokiol micelles were 5.357, 6.274, and 6.746 μg/mL, respectively. The result indicated that the cytotoxicity of the honokiol micelles was lower than that of free honokiol, which was attributed to the sustained release behaviour of honokiol from honokiol micelles [233]. Further, comparing with honokiol nanoparticles, the cytotoxicity of honokiol micelles was a little lower, which might be due to the absence of organic solvent and surfactant in the honokiol micelles [233]. To increase the hydrophilicity of honokiol, Qiu et al. [234] developed an amphiphilic polymer–drug conjugate via the condensation of low molecular weight monomethoxy-poly(ethylene glycol) (MPEG)-2000 with honokiol through an ester linkage. The MPEG–honokiol (MPEG–HK) conjugate prepared formed nano-sized micelles, with a mean particle size of less than 20 nm (MPEG–HK, 360 μg·mL^−1^) in water, in which they could be well dispersed, and the results showed that only 20% of the conjugated honokiol was released in 2 h in beagle dog plasma, while in phosphate-buffered saline, the time required to reach 20% of honokiol release was >200 h [234]. Meanwhile, the inhibitory activity of the honokiol conjugate was found to be retained in vitro against LL/2 cell lines with an IC_50_ value of 10.7 μg/mL [234]. These results suggest that the polymer–drug conjugate provides a potential new approach to hydrophobic drugs, such as honokiol, in formulation design. In another study, nanomicellar honokiol (HNK-NM) with the size range of 20–40 nm was developed and compared against honokiol free drug (HNK-FD) [212]. Compared to HNK-FD, HNK-NM resulted in a significant increase in oral bioavailability. Cmax (4.06 and 3.60-fold) and AUC (6.26 and 5.83-fold) were significantly increased in comparison to oral 40 and 80 mg/kg HNK-FD, respectively, when tested in triple negative breast cancer cell lines (MDA-MB-231, MDA-MB-453, and MDA-MB-468). The anticancer effects of these formulations were also studied in BALB/c nude mice transplanted with orthotopic MDA-MB-231 cell induced xenografts [212]. After four weeks of daily oral administration of HNK-NM formulation, a significant reduction in the tumour volumes and weights compared to free drug (*p* < 0.001) treated groups was observed. Furthermore, in 25% of the mice, the treatment resulted in a complete eradication of tumours. Increased apoptosis and antiangiogenic effects were observed in HNK-NM groups compared to HNK-FD and untreated control mice [212].

Wang et al. [228] prepared paclitaxel (PTX) and honokiol (HK) combination methoxy poly(ethylene glycol) poly(caprolactone) micelles (P–H/M) via the solid dispersion method against breast cancer (4T1). The particle size of P–H/M was 28.7 ± 2.5 nm and spherical in shape. Both the cytotoxicity and the cellular uptake of P–H/M were increased in 4T1 cells, and P–H/M induced more apoptosis than PTX-loaded micelles or HK-loaded micelles. Furthermore, the antitumor effect of P–H/M was significantly improved compared with PTX-loaded micelles or HK-loaded micelles in vivo (Female Balb/c mice and female Balb/c nude mice treated with intravenous injection) [31,228]. P–H/M were more effective in inhibiting tumour proliferation, inducing tumour apoptosis, and decreasing the density of microvasculature accumulated more in tumour tissues compared to the free drug. After that, Wang et al. [235] developed paclitaxel (PTX) and honokiol (HNK) which are co-encapsulated into pH-sensitive polymeric micelles based on poly(2-ethyl-2-oxazoline)-poly(D,L-lactide) (PEOz-PLA). Results showed efficient inhibition of tumour metastasis by dual drug-loaded PEOz-PLA micelles in vitro anti-invasion and anti-migration assessment in MDA-MB-231 cells and in vivo in nude mice [235]. The suppression of MDR and metastasis by the micelles was assigned to the synergistic effects of pH-triggered drug release and HNK/PEOz-PLA-aroused P-gp inhibition, and pH-triggered drug release and PTX/HNK-aroused MMPs inhibition, respectively. After that, Wang et al. [236] proceeded to modify the paclitaxel plus honokiol micelles with dequalinium and tested it in non-small-cell lung cancer. When tested on Lewis lung tumour (LLT) cells, the polymeric micelles show powerful cytotoxicity, effective suppression on vasculogenic mimicry (VM) channels and tumour metastasis, as well as the activation of apoptotic enzymes caspase-3 and caspase-9, and down-regulation of FAK, PI3K, MMP-2, and MMP-9 [236]. In vivo assays (C57BL/6 mice treated through intravenous injection) indicated that polymeric micelles could increase the selective accumulation of chemotherapeutic drugs at tumour sites and showed a conspicuous anti-tumour efficacy [236].

For liposomes loaded with honokiol, Luo et al. [73] created liposomal honokiol and tested it on cisplatin-sensitive (A2780s) and -resistant (A2780cp) human ovarian cancer models. The administration of liposomal honokiol resulted in significant inhibition (84–88% maximum inhibition relative to controls) in the growth of A2780s and A2780cp tumour xenografts and prolonged the survival of the treated mice (treated twice weekly with intravenous administration) [73]. These anti-tumour responses were associated with marked increases in tumour apoptosis, and reductions in intratumoural microvessel density. Jiang et al. [223] incorporated honokiol in combination with cisplatin in the liposomes and tested it in A549 lung cancer xenograft nude mice model through intraperitoneal administration. This combination effectively suppressed tumour growth and significantly increased life span of treated mice compared to liposomal honokiol alone [223]. A similar result was observed in murine CT26 colon cancer models, where the systemic administration of liposomal honokiol with cisplatin resulted in the inhibition of subcutaneous tumour growth beyond the effects observed with either liposomal honokiol or cisplatin alone due to elevated levels of apoptosis and reduced endothelial cell density significantly [237]. In a recent study, hyaluronic acid (HA) modified daunorubicin plus honokiol cationic liposomes were prepared and characterised for the treatment of breast cancer by eliminating vasculogenic mimicry (VM) [238]. Studies found that the HA modified daunorubicin plus honokiol cationic liposomes enhanced the cellular uptake and destroyed VM channels. In addition, these liposomes prolonged their circulation time in the blood, and significantly accumulated at the tumour site to maximise its anticancer efficacy.

## 9. Future Perspective

Up to date, many in vitro and in vivo studies have identified the protective effects of honokiol in various types of cancers. However, the exact anticancer mechanism of honokiol is still insufficiently elucidated, especially its application in treating human cancer clinically. Since honokiol is being extensively metabolised in the body into different metabolites, it is vital to recognise the different types of metabolites circulating in the body in order to gain a better insight into the fate of honokiol after administration. The characterisation of honokiol metabolites would enable a better understanding of the overall bioactivity of honokiol as well as to determine the relationship between the bioactivity of the core molecule and its metabolites circulating within the target tissue. Moreover, future studies could focus on improving the methods used for in vitro studies to mimic more favourable in vivo conditions by considering the actual metabolites detected and concentrations found in the respective cancer tissues in order to better understand the mode of action of honokiol in cancer. Apart from that, it is essential to study the anticancer properties of the derivatives of honokiol as very few studies have been performed on the derivatives. It is important to study its derivatives as they might have improved and enhanced anticancer properties due to the change in structures and functional groups.

In short, more research can be done to confirm the anticancer properties of honokiol in more detail in order to come up with a safe and effective dosage to be used in chemoprevention and chemotherapy. Furthermore, more research can be done on the metabolism of honokiol via different routes of administration to find out the most effective route of administration for different types of cancer. The pre-formulation as well as formulation of honokiol can also be developed to prepare the transition of honokiol from pre-clinical to clinical studies in the future.

## 10. Conclusions

For centuries, researchers have been searching for strategies to control cancer progression through different approaches. Honokiol is a potential natural compound that exerts multiple effects on different cellular processes in various cancer models. Honokiol has been shown to regulate cell cycle arrest, induction of apoptosis, necrosis, and autophagy, as well as the inhibition of metastasis and angiogenesis through various signalling pathways. In addition, its effects are also validated in several in vivo studies with promising results where it can inhibit tumour growth and prolong survival in mouse cancer models. Current efforts are focusing on developing numerous drug delivery systems to improve the pharmacological, pharmacokinetics, and pharmacodynamic properties of honokiol. This review concludes that honokiol may be considered as a potential candidate for anticancer drug development.

## Figures and Tables

**Figure 2 cancers-12-00048-f002:**
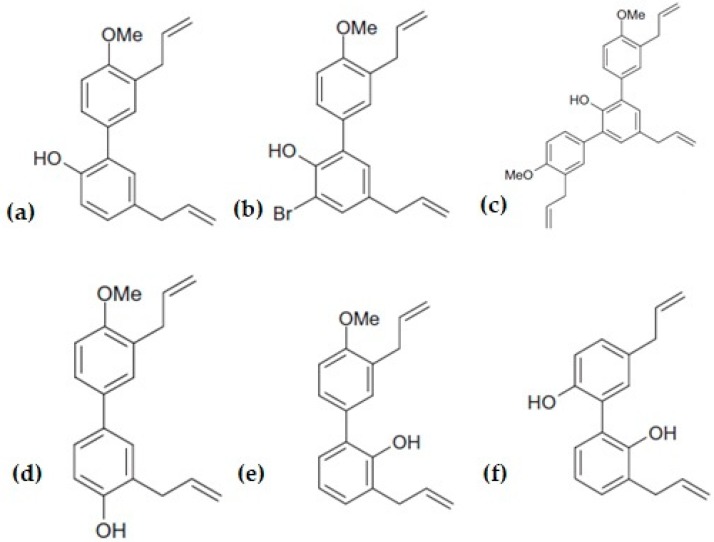
The structure of honokiol analogues. (**a**): 3,5′-Diallyl-2′-hydroxyl-4-methoxy-1,1′-biphenyl; (**b**): 3′-Bromo-3,5′-di-allyl-2′-hydroxyl-4-methoxy-1,1′-biphenyl; (**c**): 2,6-Di-(4′-methoxy-3′-allylphenyl)-1-phenol; (**d**): 3,3′-Diallyl-4-methoxy-4′-hydroxy-1,1′-biphenyl; (**e**): 3,3′Diallyl-2′-hydroxyl-4-methoxy-1,1′-biphenyl; (**f**): 3′,5-Diallyl-2,2′-di-hydroxy-1,1′-biphenyl [24].

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
