# Peer review of "Honokiol: A Review of Its Anticancer Potential and Mechanisms"

_cancers, 2019, doi:10.3390/cancers12010048_

Round 1
Reviewer 1 Report
The present manuscript is a comprehensive collection of available studies on the anti-cancer effects of Honokiol. The present manuscript is a well-designed and well executed study.
I have some minor comments.
Ong et al are trying the convey the mechanism of action of Honokiol as a single very long sentence in the abstract section from line 17 to 24. Please sub divide this sentence into short sentences so that the message can be communicated more effectively to the readers.
The authors should provide a mechanism of action highlighting the anti-cancer effect of Honokiol as a separate figure.
I would also like to suggest that they authors provide a separate figure on the effect of Honokiol on various signalling pathways.
There are multiple typo errors in the manuscript which needs to be addressed.
Author Response
Dear Reviewer,
Thank you for your input and suggestions for the manuscript. Please find our response below:
We have amended the abstract section from line 17 to 24. It was sub-divided into a few short sentences to enhance the clarity of the abstract We have included a graphical abstract (a separate figure) which highlight Honokiol's mechanisms of action and targeted signaling pathways We have addressed and corrected the typo errors in the paperReviewer 2 Report
Dear Chon et al.,
This review article has given an overview of the bio-active compound Honokiol and its anticancer potential.
This is an incredible analysis of published articles from 1994 – 2019, using many keywords in cancer biology and summarized the anti-cancer potential of Honokiol and its signaling based on in vitro and in vivo.
I recommend this review article for publication.
Author Response
Dear Reviewer,
Thank you for recommending our manuscript for publication.
Reviewer 3 Report
This manuscript provides a comprehensive review of honokiol as a potential anticancer therapeutic, including mechanisms, pharmacokinetics, and drug delivery. Generally it is a well-written manuscript but several grammatical errors throughout the manuscript should be corrected. Other minor comments are:
Review articles on honokiol have been published previously. It would be suggested to mention those reviews in the introduction and briefly state how this review will distinguish from other reviews. Line 209: “This subsection will further discuss on the mechanism ….”. It would be suggested to start a new section – section 5. Under this subsection, the different mechanisms will be discussed. Thus the currently sections 4.3-4.7 will become 5.1-5.5. Examples of grammatical errors: line 129, “will be produced”; line 131, “has greatly improve”; line 231, “Evidence from other studies also shown that honokiol disrupt”, etc.Author Response
Dear Reviewer,
Thank you for your input and suggestions for the manuscript. Please find our response below:
We have corrected the typo and grammar errors in the manuscript We have included an additional line in the introduction to state the aims of the review paper and how it differentiates from other reviews We have moved the mechanisms of actions subsection to Section 5 The grammar errors mentioned in Line 129, 131 and 231 were amended accordingly